# INSTRUCTMOLE: INSTRUCTION-GUIDED MIXTURE OF LOW-RANK EXPERTS FOR MULTI-CONDITIONAL IMAGE GENERATION

## ABSTRACT

Parameter-Efficient Fine-Tuning of Diffusion Transformers (DiTs) for diverse, multi-conditional tasks often suffers from task interference when using monolithic adapters like LoRA. The Mixture of Low-rank Experts (MoLE) architecture offers a modular solution, but its potential is usually limited by routing policies that operate at a token level. Such local routing can conflict with the global nature of user instructions, leading to artifacts like spatial fragmentation and semantic drift in complex image generation tasks. To address these limitations, we introduce InstructMoLE, a novel framework that employs an Instruction-Guided Mixture of Low-Rank Experts. Instead of per-token routing, InstructMoLE utilizes a global routing signal, Instruction-Guided Routing (IGR), derived from the user's comprehensive instruction. This ensures that a single, coherently chosen expert council is applied uniformly across all input tokens, preserving the global semantics and structural integrity of the generation process. To complement this, we introduce an output-space orthogonality loss, which promotes expert functional diversity and mitigates representational collapse. Extensive experiments demonstrate that InstructMoLE significantly outperforms existing LoRA adapters and MoLE variants across challenging multi-conditional generation benchmarks. Our work presents a robust and generalizable framework for instruction-driven fine-tuning of generative models, enabling superior compositional control and fidelity to user intent.

## 1 INTRODUCTION

The advent of powerful, open-source Diffusion Transformers (DiTs) (Peebles & Xie, 2023; Esser et al., 2024; Labs, 2024) has unlocked unprecedented capabilities in generative AI, fueling a demand for highly specialized and compositional functionalities, from multi-subject composition to personalized content creation (Labs et al., 2025; Wu et al., 2025a;b; Liu et al., 2025a). Parameter-Efficient Fine-Tuning (PEFT), particularly Low-Rank Adaptation (LoRA) (Hu et al., 2022), has become the de facto standard for such customization. However, LoRA's monolithic update structure conflicts with the demands of multi-task fine-tuning, leading to catastrophic forgetting as different task objectives interfere (Biderman et al., 2024; Han et al., 2024).

The Mixture-of-Experts (MoE) architecture offers a structured solution to this interference problem. As theoretically grounded by (Li et al., 2025), MoE mitigates catastrophic forgetting by diversifying its experts to specialize in different tasks, which in turn helps to establish, or at least preserve, task-specific expert circuits and balance the loads across them. In the domain of language and multi-modal models, MoE has been extensively studied, leading to established, often task-aware, routing strategies (Wu et al.; Liu et al., 2025c; Li et al., 2024; Gou et al., 2023; Chen et al.; Wu et al., 2025c; Sun et al., 2025; Fei et al., 2024).

In contrast, its application to diffusion transformers for multi-conditional image generation (e.g. image editing, multi-subject driven generation) remains comparatively underexplored. Initial works like ICEdit (Zhang et al., 2025b) have adopted conventional token-level routing mechanisms without specializing them for this new problem context. This reliance on an ill-suited, token-level routing mechanism reveals a misalignment with the nature of image generation instructions. A user's in-

struction for compositional generation, for instance, to create a scene with "*A baby crawling on the grass, a white horse grazing nearby, and a football helmet*" (in Figure 2), establishes a complex set of global semantic relationships. In contrast, token-level routing delegates expert selection to each local image patch independently. This flawed, uncoordinated decision-making process is the primary cause of critical failures, often resulting in subjects that are incoherently rendered, spatially fragmented, or have their specified relationships and attributes ignored entirely.

To rectify this misalignment, we propose InstructMoLE, a MoLE framework for multi-conditional image generation built upon a novel, globally consistent routing policy. The core of our framework is Instruction-Guided Routing (IGR), which conditions expert selection entirely on the global semantics of the user's textual instruction.This mechanism ensures that for any given layer of the model, a single, unified "expert council" is chosen based on the instruction and broadcast to all spatial locations within that layer. This enforces processing consistency at each stage of generation, while critically allowing the model to recruit different sets of specialized experts across different layers, tailoring the computation to the varying levels of abstraction. However, the effectiveness of this globally-applied council hinges on the functional diversity of its constituent experts. To ensure this diversity and prevent representational collapse, we complement IGR with a novel output-space orthogonality loss. This regularizer encourages the learned experts to occupy distinct functional roles, thereby maximizing the compositional power of the selected council. Our main contributions are:

- We identify and address a critical challenge in applying MoLE to instruction-based editing and generation: the inherent mismatch between local, token-level routing policies and the global, semantic scope of user instructions. To address this, we propose InstructMoLE, a framework centered on Instruction-Guided Routing (IGR), which aligns expert selection with the holistic intent of the instruction.

- We introduce a novel output-space orthogonality regularizer to explicitly promote functional diversity among experts. This technique complements standard load-balancing losses by directly penalizing representational redundancy in the expert outputs, thereby mitigating expert collapse and improving the model's compositional control.

- We provide extensive empirical validation across multiple challenging benchmarks for multi-conditional image generation. Our results demonstrate that InstructMoLE achieves state-of-the-art performance, particularly in tasks demanding high compositional fidelity and adherence to complex spatial relationships. This directly validates the superiority of our global, instruction-guided approach over traditional token-level routing policies. These findings validate that instruction-guided routing is a more effective and principled approach for multi-conditional generation tasks.

## 2 RELATED WORK

### 2.1 CONDITIONAL GENERATION WITH DIFFUSION TRANSFORMERS.

The advent of Diffusion Transformers (DiTs) has marked a new era for generative modeling, demonstrating remarkable scalability and performance (Peebles & Xie, 2023; Esser et al., 2024). A significant line of research has focused on enhancing their controllability for complex, instruction-driven tasks. Methods like DreamO (Mou et al., 2025) and In-Context Edit (Zhang et al., 2025b) have enabled sophisticated image customization and editing based on diverse user inputs. Others have introduced lightweight modules for flexible conditioning (Zhang et al., 2025a; Jiang et al., 2025) or tackled data bottlenecks in multi-subject scenarios (Wu et al., 2025d). While these works significantly advance the state of conditional generation, they typically rely on a monolithic adaptation strategy, such as a single LoRA (Hu et al., 2022). This approach struggles when tasked with mastering multiple, potentially conflicting skills simultaneously, creating a clear need for more dynamic, modular adaptation methods.

### 2.2 MIXTURE-OF-EXPERTS FOR PARAMETER-EFFICIENT FINE-TUNING.

The Mixture-of-Experts (MoE) architecture, particularly in its low-rank form (MoLE), offers a powerful solution to the limitations of monolithic fine-tuning. By allocating specialized LoRA experts

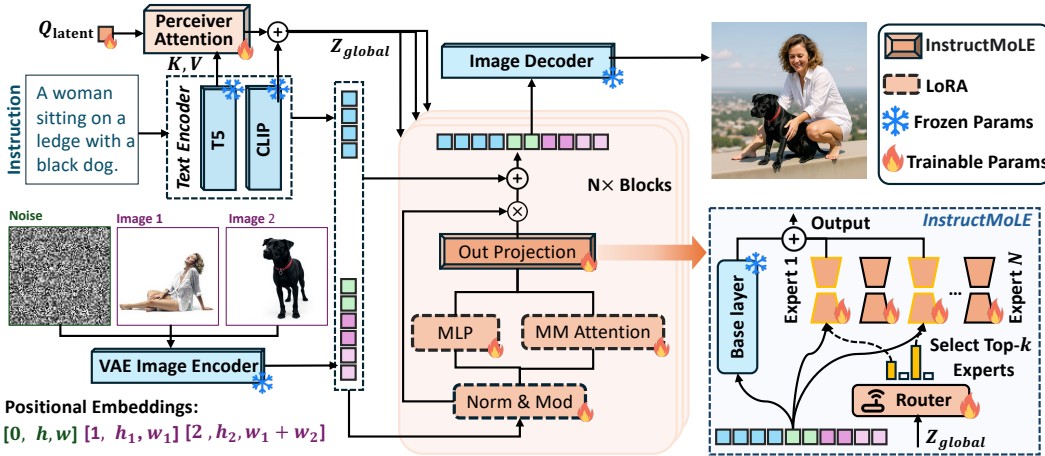

Figure 1: Illustration of the InstructMoLE framework. A global signal, $\mathbf{Z}_{\text{global}}$, is distilled from the user's instruction to guide a Router. The Router selects a single, consistent set of LoRA experts, which is then applied to all input tokens.

for different functions, MoLE can mitigate task interference and enhance model capacity efficiently. This paradigm has been extensively and successfully explored in language and multi-modal models for multi-task learning and instruction tuning (Dou et al., 2024; Gou et al., 2023; Chen et al.).Its application to vision, however, is more nascent. While initial works have validated the potential of MoLE for tasks like controllable visual effect generation (Mao et al., 2025), a well-designed routing policy that determines which experts to activate for a given input is crucial for fully harnessing the power of this architecture.

### 2.3 MoE Design: Routing Policies and Expert Diversity.

The design of an effective MoE system centers on two core challenges: the routing policy and the promotion of expert diversity. The predominant routing approach, inherited from language models, is token-level routing, where each token independently selects experts (Shazeer et al., 2017; Zhang et al., 2025b). While advanced variants like Expert Choice routing improve dynamics (Sun et al.), they inherit the same fundamental limitation: local, per-token decision-making. This is misaligned with the holistic intent of user instructions in image generation. A second, orthogonal challenge is the common issue of "expert collapse", where experts become functionally redundant. Standard load-balancing only encourages uniform utilization, not functional diversity. Our work addresses these dual challenges head-on. We introduce Instruction-Guided Routing (IGR), a global policy that aligns expert selection with the user's instruction, and complement it with an output-space orthogonality loss that directly enforces functional diversity among experts.

## 3 Methodology

We present InstructMoLE, a Mixture of Low-rank Experts (MoLE) framework for multi-conditional image generation, illustrated in Figure 1. The framework champions a singular, top-down routing principle: a unified Instruction-Guided Routing (IGR) policy is applied consistently across all model layers. This is achieved through a Perceiver-style signal distillation for robust global guidance and an output-space orthogonality loss to maintain functional diversity among experts.

### 3.1 InstructMoLE Architecture

Standard LoRA augments a frozen linear layer weight $\mathbf{W}_0 \in \mathbb{R}^{D_{\text{in}} \times D_{\text{out}}}$ with a single, static low-rank update. We replace this static update with a dynamic, sparse mixture of $N$ expert LoRA modules $\{\mathbf{E}_i\}_{i=1}^{N}$. Each expert $i$ consists of a down-projection matrix $\mathbf{W}_{\text{A}}^i \in \mathbb{R}^{D_{\text{in}} \times r}$ and an up-projection matrix $\mathbf{W}_{\text{B}}^i \in \mathbb{R}^{r \times D_{\text{out}}}$, where $r \ll D_{\text{in}}, D_{\text{out}}$ is the rank. The function for expert $i$ is defined as

$\mathbf{E}_i(\mathbf{X}) = \mathbf{X}\mathbf{W}_{\mathrm{A}}^i \mathbf{W}_{\mathrm{B}}^i$. The output of a MoLE layer is therefore:

$$\mathrm{MoLE}(\mathbf{X}) = \mathbf{X}\mathbf{W}_0 + \sum_{i=1}^{N} g_i(\cdot)\mathbf{E}_i(\mathbf{X}) \tag{1}$$

where the gating weights $g_i(\cdot)$ are determined by a routing network; $\mathbf{X} \in \mathbb{R}^{B \times L \times D_{\mathrm{in}}}$ is the input tensor with batch size $B$ and sequence length $L$.

**Token-level vs. Instance-level Routing.** Conventional MoE models employ *token-level* routing, where each of the $B \times L$ tokens independently selects experts from its local hidden state (Fedus et al., 2022; Shazeer et al., 2017; Yuan et al.; Sun et al.; Zhang et al., 2025b), resulting in a dense logits tensor of shape $\mathbb{R}^{B \times L \times N}$. The uncoordinated, per-token decisions create a tension with the inherently global nature of many instructions, leading to critical failures: (i) spatial fragmentation, yielding region-wise style and hue discontinuities; (ii) amplification of high-frequency noise through routing jitter; and (iii) weakened adherence to the user instruction.

To counteract these failure modes, we introduce an *instance-level* policy that enforces a globally consistent expert assignment at each layer. Instead of fragmented per-token decisions, our router is guided by the global semantics of the user instruction to compute a *unified expert council* for the entire instance. This council, represented by a compact logits tensor of shape $\mathbb{R}^{B \times 1 \times N}$, is then broadcast identically to all $L$ tokens within that layer. We term this mechanism Instruction-Guided Routing. By ensuring that a single, unified expert council is applied to all input tokens within a given layer, IGR directly prevents fragmentation and routing jitter, thereby preserving the semantic and structural integrity of the image.

### 3.1.1 INSTRUCTION-GUIDED ROUTING (IGR) POLICY

**Global Routing Signal ($\mathbf{Z}_{\mathbf{global}}$).** The IGR policy is conditioned on a global signal, $\mathbf{Z}_{\mathrm{global}}$, which is distilled from the editing instruction, $\mathbf{I_c}$. To construct a signal that is both semantically robust and compositionally aware, we fuse two distinct text representations. Given an $\mathbf{I_c}$, a T5 encoder produces token-level features $\mathbf{H}_{\mathrm{inst}} \in \mathbb{R}^{B \times L_{\mathrm{inst}} \times D_{\mathrm{inst}}}$, where $B$ is the batch size, $L_{\mathrm{inst}}$ is the instruction length, and $D_{\mathrm{inst}}$ is the T5 embedding dimension. Concurrently, a CLIP encoder provides a pooled embedding $\mathrm{CLIP}(\mathbf{I_c}) \in \mathbb{R}^{B \times D}$. While the pooled embedding offers a strong semantic anchor, it can overlook crucial compositional nuance. For example, in the instruction "Change the woman's dress to red and the man's shirt to blue", a pooled embedding might average "red" and "blue", losing the critical association between the colors and their respective subjects. Conversely, token-level features, if naively averaged, risk diluting these key semantics.

We therefore employ a **Perceiver-style attentional bottleneck** (Jaegle et al., 2021) to distill the rich, token-level information from $\mathbf{H}_{\mathrm{inst}}$ into a compact summary before fusing it with the CLIP embedding. Concretely, we first project the T5 tokens into the $D$-dimensional CLIP space using a linear layer $\mathbf{W}_{\mathrm{in}} \in \mathbb{R}^{D_{\mathrm{inst}} \times D}$. A single, learnable latent query, $\mathbf{Q}_{\mathrm{latent}} \in \mathbb{R}^{1 \times 1 \times D}$, is then used to iteratively query the projected instruction tokens via $S = 2$ layers of Perceiver-style attention:

$$\tilde{\mathbf{X}} = \mathbf{H}_{\mathrm{inst}}\mathbf{W}_{\mathrm{in}}, \quad \mathbf{L}^{(0)} = \mathrm{tile}_B(\mathbf{Q}_{\mathrm{latent}}),$$
$$\mathbf{L}^{(s)} = \mathbf{L}^{(s-1)} + \mathrm{PerceiverAttn}(Q = \mathbf{L}^{(s-1)}, K = \tilde{\mathbf{X}}, V = \tilde{\mathbf{X}}), \tag{2}$$

where $\mathrm{tile}_B(\cdot)$ repeats the latent query $B$ times. The final latent, $\mathbf{L}^{(S)}$, now serves as a distilled summary of the compositional details present in the T5 features. This summary is projected with $\mathbf{W}_{\mathrm{out}} \in \mathbb{R}^{D \times D}$, normalized, and then additively fused with the CLIP embedding:

$$\mathbf{Z}_{\mathrm{global}} = \underbrace{\mathrm{LayerNorm}(\mathbf{L}^{(S)}\mathbf{W}_{\mathrm{out}})}_{\text{distilled compositional summary}} + \underbrace{\mathrm{CLIP}(\mathbf{I_c})}_{\text{holistic semantics}}. \tag{3}$$

This design allows the distilled term to contribute fine-grained specificity from the instruction, while the CLIP embedding provides a robust, holistic semantic anchor, together forming a balanced and powerful global routing signal.

**IGR Gating Mechanism.** The IGR gating mechanism is instantiated independently at each MoLE layer. We formalize the computation by first describing the process for a single instance

$b \in \{1, \ldots, B\}$ within a batch, which corresponds to the input tensor slice $\mathbf{X}_b \in \mathbb{R}^{L \times D_{\text{in}}}$. The per-instance outputs, $\mathbf{Y}_b$, are subsequently stacked to form the full batch output $\mathbf{Y}$.

At a given $l$-th layer, the gating network $\mathcal{G}_l : \mathbb{R}^D \to \mathbb{R}^N$ generates a logit vector over the $N$ experts from a shared global instruction signal, $\mathbf{Z}_{\text{global},b}$. The Top-$k$ experts are then selected:

$$(\mathcal{I}_b, \mathbf{w}_b) = \text{Top-}k(\text{Softmax}(\mathcal{G}_l(\mathbf{Z}_{\text{global},b})), k). \tag{4}$$

Here, $\mathcal{I}_b \subset \{1, \ldots, N\}$ is the set of indices for the Top-$k$ experts, and $\mathbf{w}_b \in \mathbb{R}^k$ contains their associated, unnormalized Softmax probabilities. $\mathcal{I}_b$ and $\mathbf{w}_b$ is then broadcast across all $L$ token positions in the sequence. The final output for the $l$-th layer, $\mathbf{Y}_b$, is computed by summing the output of a shared linear layer with the weighted sum of the selected expert outputs:

$$\mathbf{Y}_b = \mathbf{X}_b \mathbf{W}_0 + \sum_{j=1}^{k} w_{b,j} \cdot \mathbf{E}_{\mathcal{I}_b[j]}(\mathbf{X}_b), \tag{5}$$

where $w_{b,j}$ is the weight for the $j$-th selected expert, whose global index is $\mathcal{I}_b[j]$), and $\mathbf{E}_{\mathcal{I}_b[j]}$ denotes the corresponding low-rank expert.

## 3.2 TRAINING OBJECTIVES

Our training objective is designed to achieve two goals simultaneously: high-fidelity image generation and the effective, diverse utilization of the expert pool. This is realized through a composite loss function, where the $\lambda$ terms are hyper-parameters:

$$\mathcal{L}_{\text{total}} = \mathcal{L}_{\text{flow}} + \lambda_{\text{aux}} \mathcal{L}_{\text{aux}} + \lambda_{\text{ortho}} \mathcal{L}_{\text{ortho}}. \tag{6}$$

**Generation Accuracy ($\mathcal{L}_{\text{flow}}$).** The primary objective, $\mathcal{L}_{\text{flow}}$, is the standard flow-matching or diffusion loss of the MM-DiT backbone. This ensures the model accurately synthesizes images that are well aligned with the user's instruction.

**Expert Load Balancing ($\mathcal{L}_{\text{aux}}$).** Following the standard practice for training MoEs (Shazeer et al., 2017), we employ an auxiliary load-balancing loss, $\mathcal{L}_{\text{aux}}$, to prevent the model from consistently favoring only a few experts. This loss is a function of two quantities calculated over a batch: $f_i$, the fraction of tokens routed to expert $i$, and $p_i$, the mean routing probability for that expert:

$$\mathcal{L}_{\text{aux}} = N \sum_{i=1}^{N} f_i \cdot p_i, \tag{7}$$

where $N$ is the total number of experts.

**Output-space Orthogonality Loss for Functional Diversity ($\mathcal{L}_{\text{ortho}}$).** A critical failure mode in Mixture-of-Experts (MoE) models is *expert collapse* (Fedus et al., 2022), where distinct experts converge to functionally redundant solutions, thereby nullifying the benefits of the mixture. The standard mitigation, an auxiliary load-balancing loss, only encourages that experts are utilized with similar frequency but provides no explicit mechanism to ensure their functional diversity.

To address this limitation directly, we introduce an **orthogonality loss** that penalizes the functional similarity between pairs of experts. Our approach operates on the raw, pre-gating outputs. For a given input batch $\mathbf{X}$, we first compute the output of every expert, yielding a set of tensors $\{\mathbf{Y}_1, \ldots, \mathbf{Y}_N\}$, where each $\mathbf{Y}_i = \mathbf{E}_i(\mathbf{X}) \in \mathbb{R}^{B \times L \times D_{\text{out}}}$. To measure the functional similarity between any two experts $i$ and $j$, we flatten their respective output tensors into high-dimensional vectors, $\mathbf{v}_i = \text{vec}(\mathbf{Y}_i)$ and $\mathbf{v}_j = \text{vec}(\mathbf{Y}_j)$.

The orthogonality loss, $\mathcal{L}_{\text{ortho}}$, is then defined as the mean of the squared cosine similarities over all unique pairs of these expert output vectors:

$$\mathcal{L}_{\text{ortho}} = \frac{1}{N(N-1)} \sum_{i \neq j} \left( \frac{\mathbf{v}_i \cdot \mathbf{v}_j}{\|\mathbf{v}_i\|_2 \|\mathbf{v}_j\|_2} \right)^2. \tag{8}$$

Minimizing this objective incentivizes the vectors $\{\mathbf{v}_i\}_{i=1}^{N}$ to become mutually orthogonal. This, in turn, forces the expert functions $\{\mathbf{E}_i\}_{i=1}^{N}$ to learn distinct and complementary representations,

Figure 2: Qualitative comparison with state-of-the-art models.

effectively preventing expert collapse. In practice, this loss is computed efficiently by calculating the squared off-diagonal elements of the Gram matrix formed by the L2-normalized vectors $\{\mathbf{v}_i/\|\mathbf{v}_i\|_2\}_{i=1}^{N}$. This is crucial for unlocking the full specialization capacity of our MoLE framework.

# 4 EXPERIMENTS

## 4.1 EXPERIMENTAL SETUP

**Training Data.** Our model's versatile editing capabilities are a direct result of its comprehensive training mixture. We curate a large-scale dataset by combining publicly available sources with a vast corpus of synthesized data, designed to expose the model to a diverse spectrum of conditional inputs. As illustrated in Figure 6, this includes reference-based tasks (e.g., face swapping, style transfer, re-lighting), multi-subject compositional generation, single-image editing, and spatially controlled generation from both dense (depth, Canny maps) and pose skeleton signals. This diverse training regimen enables a single, unified model to handle a wide array of editing modalities.

**Evaluation Benchmarks.** We evaluate all models on a suite of benchmarks, each targeting a distinct capability: OmniContext (Chen & Gupta, 2025) for in-context generation, XVerseBench (Chen et al., 2025) for multiple subjects-driven generation, GEdit-EN-full (Liu et al., 2025b) for single-image editing, and MultiGen-20M (Qin et al., 2023) and COCO Pose 2017 (Lin et al., 2014) for spatially controlled generation.

In the following tables, the best results are in **bold** and the second best are underlined. ($\downarrow$: Lower is better; $\uparrow$: Higher is better).

## 4.2 MAIN RESULTS

**Implementation Details.** We fine-tune our model from the Flux.1 Kontext (dev) backbone for 100K steps on 64 NVIDIA H100 GPUs. For the MoLE layers, we use $N = 8$ total experts with $k = 4$ activated per token and an expert rank of $r = 32$. The model is optimized using AdamW with a constant learning rate of $1 \times 10^{-4}$ and a global batch size of 256 (4 per device).

**Qualitative Comparison.** Figure 2 qualitatively compares our model against state-of-the-art methods on several complex editing tasks, visually confirming its superiority. In tasks requiring

Table 1: Quantitative comparison on the OmniContext benchmark, evaluating both prompt following and identity similarity (ID-Sim), with corresponding qualitative results shown in Figure 3.

| Method | Prompt Following | ID-Sim | | | | | |
|---|---|---|---|---|---|---|---|
| | | Avg. | multi character | multi character object | scene character | scene character object | single character |
| UNO | 5.63 | 16.56 | 20.65 | 15.16 | 16.10 | 8.49 | 22.41 |
| DreamO-v1.1 | 6.10 | 15.47 | 19.30 | 9.46 | 12.41 | 8.42 | 27.77 |
| OmniGen2 | **7.58** | 23.81 | 27.52 | 18.68 | 18.20 | 12.61 | 42.02 |
| Flux.1 Kontext (dev) | 6.24 | _36.29_ | **50.08** | _31.66_ | _30.10_ | _14.97_ | _54.65_ |
| InstructMoLE | _6.75_ | **38.85** | _45.12_ | **34.03** | **34.59** | **20.04** | **60.47** |

Figure 3: Qualitative comparison of in-context generation on OmniContext benchmark.

precise spatial control like *Spatial Align* and *Try-on*, competing models often fail to respect geometric constraints, yielding misaligned objects or incorrect poses. In contrast, our model robustly adheres to both dense (depth) and sparse (pose) guidance. Our method also demonstrates a stronger grasp of compositional semantics and identity preservation, particularly in the *Multi-Subjects* task where it correctly renders all subjects and their relationships. This advantage is most pronounced in the *Swap Face* task, where InstructMoLE is the only model to produce a coherent result while others fail. This ability to disentangle and execute the spatial, semantic, and identity components of a complex instruction highlights the effectiveness of our approach.

**In-Context Generation.** As shown in Table 1, InstructMoLE achieves a new state-of-the-art on the OmniContext benchmark, attaining the highest average ID-Sim score. The benchmark evaluates instruction adherence (assessed by GPT-4.1) and identity preservation (ID-Sim) (Deng et al., 2019). While OmniGen2 scores highest in *Prompt Following*, it suffers from poor identity preservation, scoring 39% lower than InstructMoLE (Ours) on the ID-Sim average. InstructMoLE demonstrates a superior balance, outperforming the strong Flux.1 Kontext baseline on both metrics. Although Flux.1 Kontext scores highest on the *multi character* sub-metric, this is due to rendering artifacts where subjects are naively concatenated rather than composed into a coherent scene, as shown in Figure 3 and Figure 9.

Table 2: Quantitative comparison of multi-subject driven generation. Corresponding qualitative results are presented in Figure 4.

| Method | DPG | ID-Sim | IP-Sim | Avg. |
|---|---|---|---|---|
| UNO | 74.35 | 31.96 | 53.07 | 53.13 |
| DreamO-v1.1 | _89.92_ | _58.86_ | 59.93 | _69.57_ |
| OmniGen2 | **91.91** | 35.95 | _60.93_ | 62.93 |
| Flux.1 Kontext (dev) | 89.49 | 53.12 | 59.78 | 67.46 |
| InstructMoLE | 89.57 | **60.84** | **62.81** | **71.07** |

**Multiple Subjects-driven Generation.** We evaluate the model's ability to compose multiple subjects using XVerseBench (Chen et al., 2025), a benchmark comprising 20 human identities, 74 unique objects, and 45 different animal species. As detailed in Table 2, we assess performance using three key metrics: instruction adherence (DPG score (Hu et al., 2024)), human identity preservation (Face ID similarity (Deng et al., 2019)), and object fidelity (DINOv2 similarity (Oquab et al., 2024)).

Figure 4: Qualitative comparison of multi-subject driven generation on XVerse benchmark.

Figure 5: Qualitative comparison of single-image editing on GEdit-EN-full benchmark.

InstructMoLE achieves a new state-of-the-art on the benchmark. It outperforms the strong Flux.1 Kontext baseline on the average score by a significant margin of over 3.6. While its instruction adherence (DPG) is on par with other leading methods, our model establishes a decisive advantage in subject fidelity, securing the top scores in both identity preservation (ID-Sim) and object similarity (IP-Sim). More qualitative comparison results are presented in Figure 10.

**Single-image Editing.** As shown in Table 3, on the GEdit-EN-full benchmark, which evaluates performance across 11 distinct real-world editing categories by GPT-4.1, InstructMoLE again demonstrates its superior versatility by achieving the highest average score. Our model surpasses the strong Flux.1 Kontext baseline and exhibits a decisive advantage over other leading methods like OmniGen2, with an overall performance improvement of more than 13% over the latter. This strength is particularly evident in its ability to handle both global and local manipulations with high fidelity. It establishes a new state-of-the-art in fundamental tasks such as *Color Alter*, *Material Alter*, and *Replace*, while remaining highly competitive across nearly all other categories.

Table 3: Quantitative comparison of single-image editing. The benchmark comprises 11 fine-grained editing categories reflecting practical user requests, with performance for each assessed by GPT-4.1. The corresponding qualitative results shown in Figure 5.

| Model | BG Change | Color Alt. | Mat. Alt. | Motion | Portrait | Style | Add | Remove | Replace | Text | Tone | Avg |
|---|---|---|---|---|---|---|---|---|---|---|---|---|
| DreamO-v1.1 | 3.06 | 1.66 | 2.35 | 3.76 | 3.24 | 3.34 | 2.16 | 0.55 | 3.02 | 1.51 | 2.06 | 2.43 |
| ICEdit | 2.73 | 6.00 | 4.41 | 1.74 | 2.14 | 5.19 | 4.41 | 1.53 | 4.22 | 1.58 | 4.58 | 3.50 |
| OmniGen2 | 6.99 | 5.10 | 5.11 | 3.93 | 4.59 | 6.88 | 6.17 | 4.68 | 6.45 | 4.04 | 6.05 | 5.45 |
| Flux.1 Kontext (dev) | 6.99 | 7.17 | 5.60 | 3.13 | 4.29 | 6.70 | 6.90 | 6.92 | 6.27 | 5.56 | 7.14 | 6.06 |
| InstructMoLE | 7.03 | 7.46 | 5.81 | 3.29 | 4.20 | 6.79 | 7.03 | 6.87 | 6.53 | 5.43 | 7.42 | 6.17 |

## 4.3 ABLATION STUDIES

**Implementation Details.** We train all models in our ablation studies under a unified and consistent experimental protocol to ensure a fair comparison, using 8 NVIDIA H100 GPUs for 20K steps.

**Evaluation Metrics.** We conduct a comprehensive evaluation across three key domains. For *multiple subjects-driven generation* and *single-image editing*, we report scores on the XVerseBench (Chen et al., 2025), GEdit-EN-full (Zhang et al., 2024). For *spatial alignment*, we evaluate on 500 samples for each modality, randomly sampled from the evaluation splits of MultiGen-20M (Qin et al., 2023) for depth and Canny edge control, and COCO Pose 2017 (Lin et al., 2014) for human pose control. The corresponding metrics are Root Mean Squared Error (RMSE) for depth, F1 score for Canny edges, and an F1 score based on Object Keypoint Similarity (OKS) for pose.

Table 4: Ablation study on routing policies. "Standard" refers to the classic token-level Top-$k$ routing policy (Fei et al., 2024). "Standard+IGR" applies the Standard policy to the early dual-stream blocks and IGR to the later single-stream blocks. "IGR+Standard" applies the reverse configuration.

| Model | Routing Policy | Multi-Subject (↑) | Single-Subject (↑) | Depth MSE (↓) | Canny F1 (↑) | Pose F1 (↑) |
|---|---|---|---|---|---|---|
| Flux.1 Kontext (dev) | - | 63.87 | 6.06 | 63.44 | 13.72% | 0.95% |
| LoRA ($r = 256$) | - | 64.68 | 6.06 | 35.92 | 40.43% | 31.94% |
| | Standard | 64.56 | 6.07 | 38.64 | 28.06% | 19.50% |
| | Expert Choice (EC) | **65.81** | 6.07 | 34.00 | 38.55% | 31.47% |
| MoLE | Expert Race (ER) | 55.83 | 6.11 | 102.67 | 16.73% | 0.00% |
| ($r = 32, N = 8, k = 4$) | IGR+Standard | 65.17 | **6.15** | 33.43 | 38.33% | 33.77% |
| | Standard+IGR | 64.86 | 6.07 | 36.32 | 33.33% | 9.57% |
| | IGR (ours) | 65.68 | **6.15** | **33.34** | **41.51%** | **40.97%** |

Table 5: Ablation study of the IGR signal and the orthogonality loss.

| IGR Signal | Orthogonality Loss | Multi-Subject (↑) | Single-Subject (↑) | Depth MSE (↓) | Canny F1 (↑) | Pose F1 (↑) |
|---|---|---|---|---|---|---|
| $CLIP(\mathbf{I_c})$ | w/o | 64.54 | 6.12 | 35.67 | 38.12% | 36.74% |
| $\mathbf{Z}_{global}$ | w/o | 65.65 | 5.97 | 33.74 | 38.92% | 37.77% |
| $\mathbf{Z}_{global}$ | w/ | **65.68** | **6.15** | **33.34** | **41.51%** | **40.97%** |

**Routing Policy Analysis.** We compare our Instruction-Guided Routing (IGR) against state-of-the-art token-level policies in Table 4, using a high-rank LoRA as a strong baseline under a fair training parameter budget. The results reveal the inherent instability of token-level routing for this domain; the standard Top-$k$ policy degrades performance, and this failure is exacerbated by the flexible Expert Race (ER) (Sun et al.), which collapses entirely on the Pose F1-task. While the load-balanced Expert Choice (EC) (Yuan et al.) proves to be a more stable token-level alternative, it is consistently outperformed by our IGR policy. IGR's superiority demonstrates that for multi-conditional generation, enforcing a coherent, instance-level signal is more critical than the adaptive, token-wise compute allocation offered by even the strongest token-level methods.

**Analysis of IGR Components.** We conduct a fine-grained ablation on the core components of our IGR policy in Table 5. The results provide two clear validations. First, replacing the raw CLIP embedding with our fused signal, $\mathbf{Z}_{global}$, significantly enhances performance across nearly all metrics, confirming the necessity of distilling compositional details from token-level features for precise control. Second, building upon this superior signal, the addition of our orthogonality loss yields another substantial boost, particularly in spatial alignment, increasing the Canny F1 score by over 2.5 and the Pose F1 score by over 3 percentage points. This confirms that both our signal distillation and our orthogonality regularizer are critical and complementary components for achieving state-of-the-art performance.

## 5 CONCLUSION

In this work, we addressed multi-task interference in parameter-efficient diffusion models by resolving the conflict between local token-level routing and the global intent of user instructions. We introduced InstructMoLE, a framework built upon Instruction-Guided Routing (IGR). This policy enforces a globally consistent expert selection within each layer based on the user's instruction, a mechanism complemented by an output-space orthogonality loss that promotes expert diversity. Our extensive experiments show that InstructMoLE significantly outperforms LoRA and other MoLE variants on challenging multi-conditional benchmarks. Ultimately, our work validates that a global, instruction-aware routing policy is a more robust and effective paradigm for complex generative tasks, laying the groundwork for models with more faithful compositional control.

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

# A APPENDIX

## A.1 THE USE OF LARGE LANGUAGE MODELS (LLMS)

We acknowledge the use of a large language model (LLM) to aid in the writing process of this manuscript. Its application was confined to language enhancement tasks, such as proofreading for grammatical and spelling errors, rephrasing sentences to improve clarity and readability, and ensuring a consistent formal tone. The LLM served exclusively as a writing assistant.

## A.2 TRAINING DATA DETAILS

Our model's versatile capabilities are the direct result of a comprehensive, mixed training dataset, visually summarized in Figure 6. This dataset is meticulously curated to expose the model to a wide spectrum of conditional inputs, combining large-scale, self-synthesized data for complex compositional tasks with established public datasets for foundational spatial control.

**Synthesized Data Corpus.** To enable sophisticated, instruction-driven editing and composition, we generated a large-scale corpus of training examples. This corpus focuses on tasks that are poorly represented in public datasets but are crucial for real-world applications. Our synthesis pipeline covers the following key areas:

- **Reference-based Generation:** This category involves tasks that condition the output on multiple input images and a textual instruction.
  - **Face Swapping:** Training triplets consist of a source face image, a target person's image, and an instruction to transfer the facial identity.
  - **Multi-Subject Composition:** To teach complex relational reasoning, we generate scenes described by prompts involving multiple, often unrelated, subjects (e.g., *"An old man, a French Bulldog, a tuba, and a rattlesnake in a sunny park"*).
  - **Virtual Try-on:** Samples include a person, multiple clothing items, and potentially a pose skeleton, with instructions to dress the person in the specified apparel.
  - **Re-Lighting:** Data consists of a foreground subject and a new background, with the goal of seamlessly integrating the subject into the new lighting environment.
  - **Style Transfer:** We provide a content image and a style reference image (e.g., a photograph and a sketch), instructing the model to render the content in the given artistic style.
- **Single Image Editing:** We employ the **GPT-IMAGE-EDIT-1.5M** dataset (Wang et al., 2025). This corpus contains over 1.5 million high-quality samples, constructed by leveraging GPT-4o to unify and refine existing public datasets such as OmniEdit, HQ-Edit, and UltraEdit. The refinement process ensures superior visual fidelity and semantic alignment for instruction-based editing tasks.

**Public Datasets.** To build a robust foundation in controllable generation, we incorporate several large-scale, publicly available datasets. These datasets provide strong supervision for fundamental spatial and structural alignment tasks.

- **Spatial Alignment Control:** We leverage datasets that provide dense spatial conditioning maps.
  - **Depth Control:** We utilize datasets such as SubjectSpatial-200K, which provide image-depth map pairs, to train the model to generate scenes with accurate spatial layouts.
  - **Canny Edge Control:** Similarly, we use datasets with image-Canny edge pairs to enable generation from structural outlines.
  - **Pose Control:** We use the widely adopted COCO 2017 dataset (Lin et al., 2014) with its OpenPose keypoint annotations to teach the model to generate human figures conforming to specific pose skeletons.

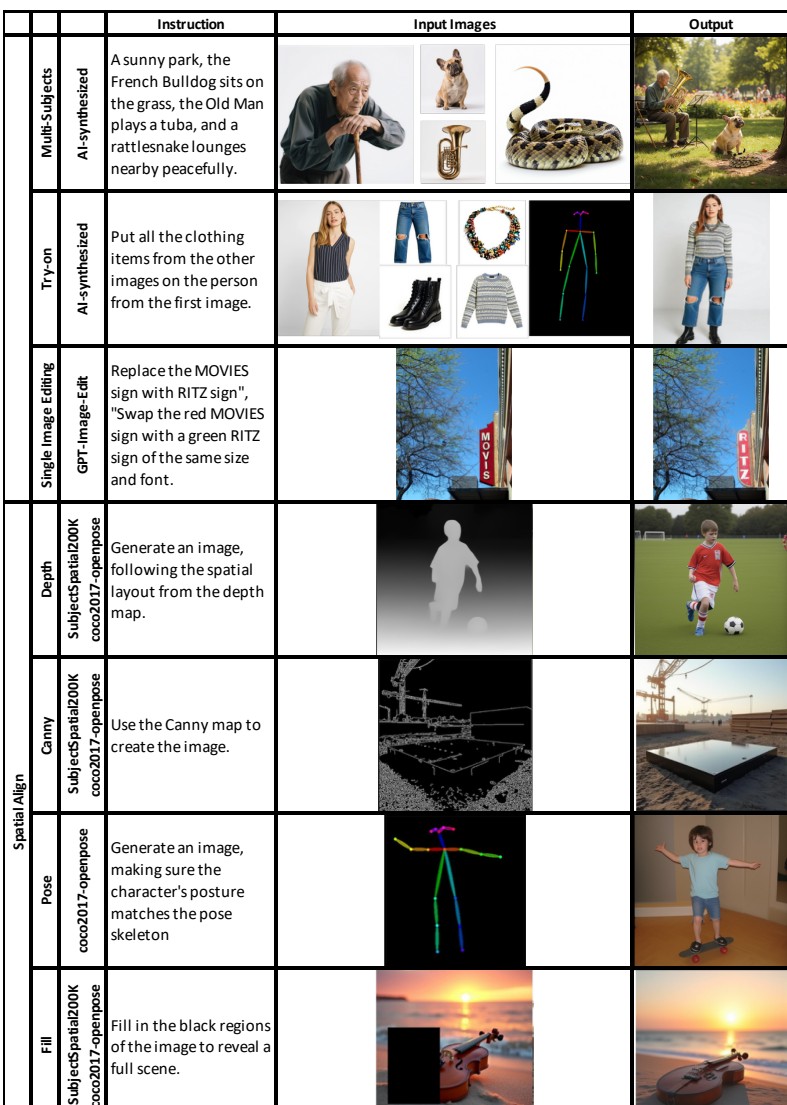

Figure 6: Training Data Details.

- **Image Inpainting (Fill):** Using datasets like SubjectSpatial-200K, we generate training examples by randomly masking regions of an image. This trains the model to fill in missing parts coherently, a crucial skill for image editing and out-painting tasks.

## A.3 ABLATION STUDY

Table 6: Ablation study on the MoLE hyper-parameter configuration.

| MoLE | | | Multi-Subject (↑) | Single-Subject (↑) | Depth MSE (↓) | Canny F1 (↑) | Pose F1 (↑) |
|---|---|---|---|---|---|---|---|
| $r$=32 | $N$=8 | $k$=2 | 64.60 | 6.11 | 35.25 | 35.10% | 36.47% |
| $r$=32 | $N$=8 | $k$=4 | **65.68** | **6.15** | **33.34** | **41.51%** | **40.97%** |
| $r$=32 | $N$=8 | $k$=6 | 64.11 | 6.10 | 33.35 | 37.53% | 37.90% |
| $r$=32 | $N$=8 | $k$=8 | 65.22 | 6.05 | 34.17 | 37.42% | 34.61% |
| $r$=64 | $N$=4 | $k$=2 | 65.06 | 6.05 | 33.66 | 38.27% | 34.77% |
| $r$=128 | $N$=2 | $k$=1 | 65.10 | **6.15** | 35.50 | 37.52% | 34.02% |

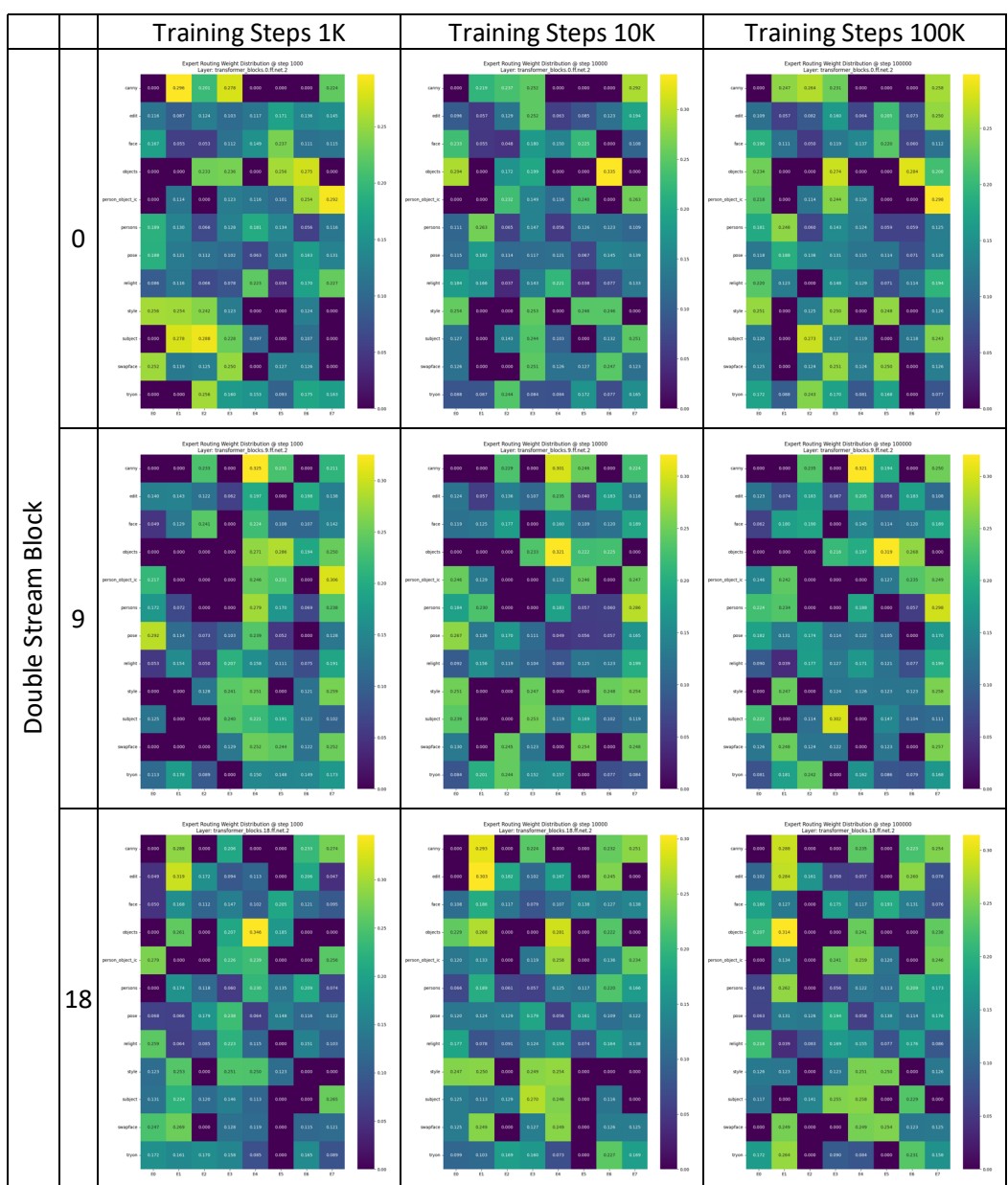

Figure 7: Visualization of the expert routing weight distributions in Double Stream Block. The numbers under the "Single Stream Block" column on the left (0, 9, 18) represent the layer index.

**Analysis of MoLE Hyper-parameters.** We analyze the MoLE configuration in Table 6 to justify our hyper-parameter choices. The results yield two key insights. First, for a fixed expert pool of $N = 8$ and rank $r = 32$, performance peaks when activating $k = 4$ experts. Activating more experts ($k > 4$), especially in the non-sparse case ($k = 8$), leads to performance degradation, highlighting the importance of sparse activation in mitigating expert interference. Second, for a fixed activation budget ($r \times k \approx 128$), our configuration with a larger pool of diverse, low-rank experts ($N = 8, r = 32$) is more effective than alternatives with fewer, high-capacity experts (e.g., $N = 4, r = 64$). This suggests that a greater diversity of specialized experts is more beneficial than individual expert capacity. These findings validate our use of the ($r = 32, N = 8, k = 4$) configuration for all main experiments.

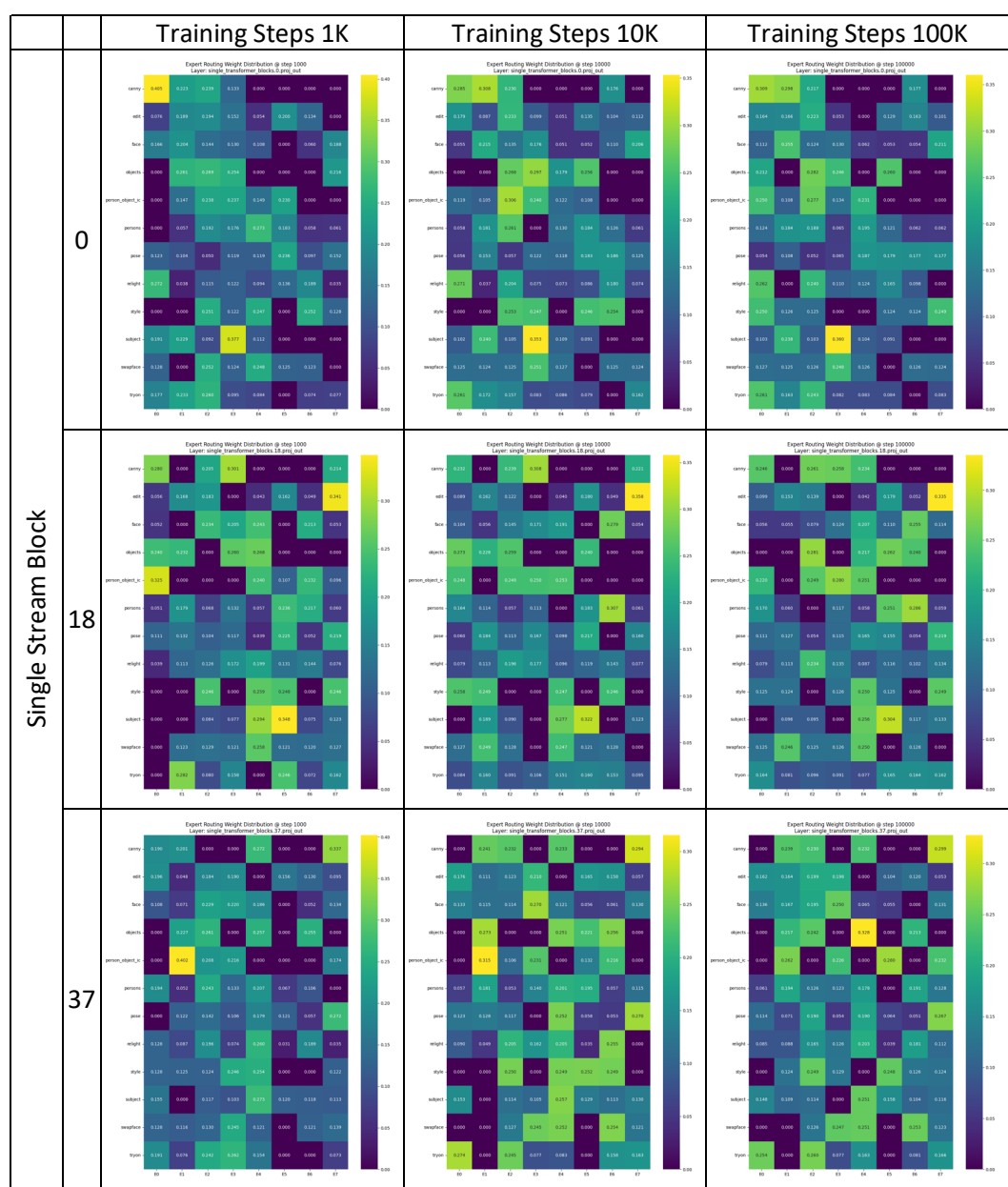

Figure 8: Visualization of the expert routing weight distributions in Single Stream Block. The numbers under the "Single Stream Block" column on the left (0, 18, 37) represent the layer index.

**Analysis of Expert Specialization Dynamics.** To investigate the learning dynamics of our MoE framework, we visualize the expert routing weight distributions at different training stages (1K, 10K, and 100K steps) and across various model depths, as shown in Figure 7 and Figure 8 The analysis reveals two key findings.

First, we observe a clear evolutionary trajectory from an initial, diffuse routing policy to a stable and highly specialized one. At 1K steps, the weights are distributed relatively evenly, indicating an early exploratory phase. By 100K steps, the distributions become notably sparse and peaked, with specific experts consistently chosen for particular task categories. This progression from a generalized to a specialized routing policy demonstrates the effective convergence of our training objectives in guiding functional disentanglement.

Second, the nature of expert specialization varies with model depth, suggesting a hierarchical division of labor. In early layers (e.g., Single Stream Block 0), experts tend to specialize in processing fundamental input types, such as distinguishing "subject" from "objects". In medial layers (e.g., Single Stream Block 18), we observe the emergence of clear, task-level specialists, with distinct experts strongly favoring tasks like "canny" or "swapface". In later layers (e.g., Single Stream Block 37), the routing pattern often becomes more distributed again, suggesting a shift from semantic task execution to a more collaborative final synthesis stage involving multiple experts.

A.4 QUALITATIVE COMPARISON

Figure 9: Qualitative comparison on OmniContext benchmark.

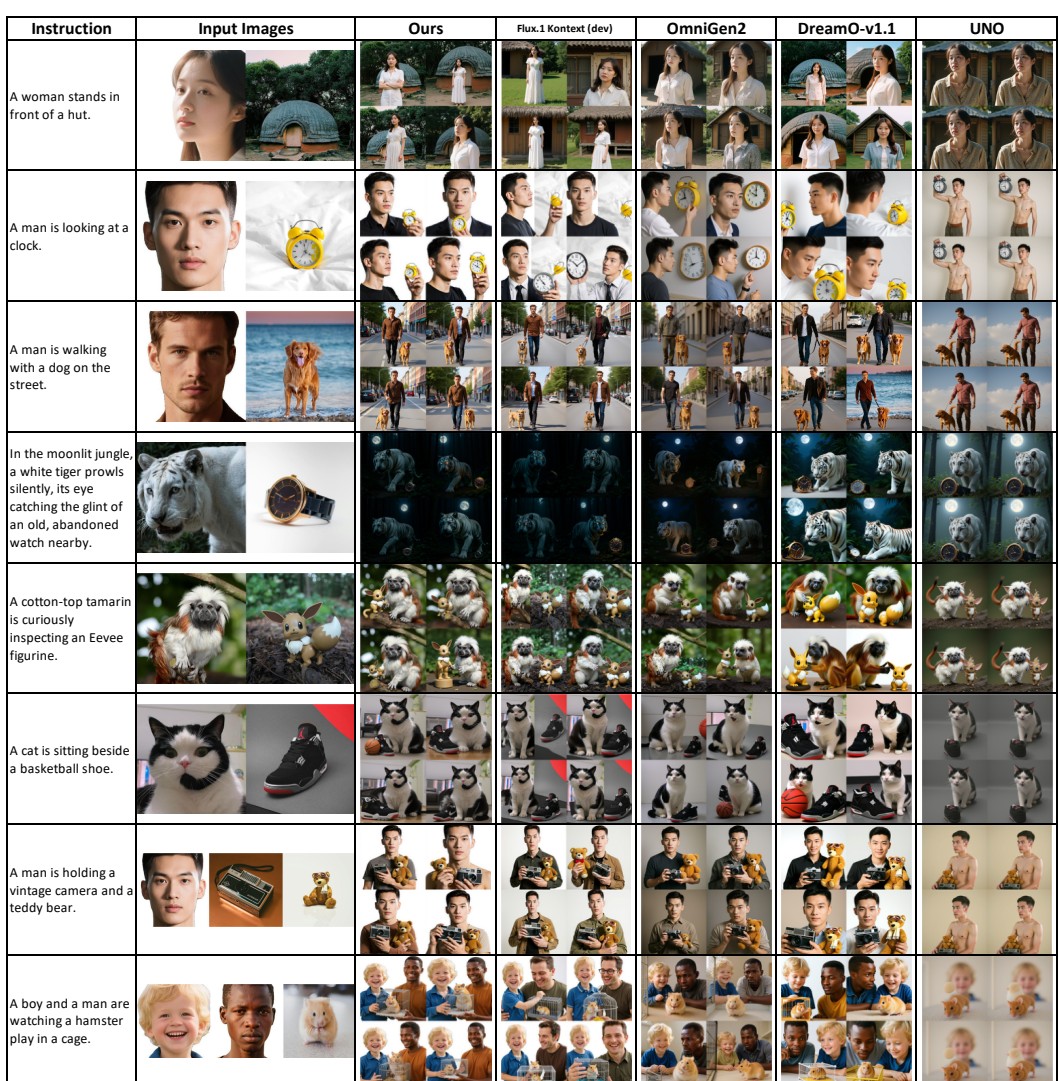

Figure 10: Qualitative comparison on Xverse benchmark.

| Instruction | Input Images | Ouput Image |
|---|---|---|
| Canny edge of the first image. Depth map of the second image. Take the a small, sleek toy racing car. in the third image as reference to generate the final image. | | |
| Generate an image by structural outline from the first image and 3D structure from the second image and take the A versatile plastic kitchen storage container. in the third image as reference, making sure the final result is Embracing the spirit of a family BBQ, it's filled with marinated meats against a backdrop of a smoky grill and lively laughter. | | |
| A cheerful sunny day at a riverside; the Cyclops peacefully reading while an alligator lounges nearby, a water lily floating gently on the water, and a windmill spinning in the background. | | |
| A bright sunny field, the Baby playing with a lily in her hand while the Old Woman plays a joyful tune on the Banjo. | | |
| Dress the person from the first image with the bootcut jeans, the denim jacket, the chelsea boots, and the wide-brim hat. | | |
| Based on the a dark purple cloak with intricate gold trim from the first image, the a shiny silver saucer with reflective surface from the second image, and the an asian woman with long hair covering her face from the third image, generate an image of the following scene: a sunny garden, the asian woman with long hair covering her face sitting on a bench, wearing the dark purple cloak, the shiny silver saucer resting beside her on the table | | |

Figure 11: Sample outputs generated by our model (InstructMoLE) from diverse, multi-modal instructions.

### A.5 QUALITATIVE ANALYSIS OF EXTREME SCENARIOS

To rigorously assess the operational boundaries of our method, we conducted stress tests on extreme scenarios involving textual noise, implicit reasoning, and high-complexity composition, as visualized in Figure 12.

**Limitations and Common Failure Modes.** Our analysis reveals a discernible threshold for robustness. In Case #1, characterized by severe typos (e.g., "Chnage the bckgrnd..."), both InstructMoLE and the baseline fail to resolve the semantic intent. This indicates that while the CLIP semantic anchor provides stability, it cannot fully compensate for T5 token embeddings when textual corruption is excessive. Similarly, in the ultra-complex Case #5, which requires simultaneous reference binding for three distinct entities ("robot," "boy," "girl"), both models struggle to correctly map identities to the generated subjects. This points to a capacity bottleneck in the underlying cross-attention mechanism of the foundation model when handling high reference loads, a limitation that persists regardless of the routing strategy.

**Superiority in Reasoning and Coherence.** Despite these boundary conditions, InstructMoLE exhibits significantly stronger capabilities in *implicit reasoning* and *atmospheric consistency*. In Case #2 ("Make it look dangerous"), our model successfully modulates the robot's expression and lowers

Figure 12: **Extreme Scenario Stress Test: Flux.1 Kontext vs. InstructMoLE.** We evaluate robustness across five edge cases spanning input noise, implicit reasoning, and complex composition. While both models share limitations under severe textual corruption (#1) and high reference load (#5), InstructMoLE demonstrates superior capabilities in abstract atmospheric inference (#2) and logical causality (#4), successfully executing implicit edits where the baseline fails.

scene lighting to align with the semantic tone, whereas the baseline output remains incongruously bright. Crucially, in Case #4 ("The person is cold"), InstructMoLE correctly infers the causal implication to generate clothing (a jacket), a logical step the baseline fails to execute. Furthermore, in multi-attribute editing (Case #3), our method achieves superior spatial and lighting integration of the inserted object.

