# OpenReview forum: "InstructMoLE: Instruction-Guided Mixture of Low-rank Experts for Multi-Conditional Image Generation"
_ICLR.cc/2026/Conference — ICLR 2026 Conference Desk Rejected Submission_

### Official Review · Reviewer_qQ3z · 2025-10-16

**Soundness:** 3
**Presentation:** 3
**Contribution:** 2
**Rating:** 6
**Confidence:** 3

**Summary:**

This paper explores the efficient fine-tuning of Diffusion Transformer (DiT) models for multi-conditional image generation tasks. The authors point out that existing Mixture-of-Experts (MoE) based methods, such as MoLE, typically employ a localized, token-level routing strategy. This conflicts with the global nature of user instructions, leading to issues such as spatial incoherence and semantic drift in the generated images.
To address this core conflict, the paper proposes a novel framework named InstructMoLE. Its core contribution is an Instruction-Guided Routing (IGR) mechanism. Unlike per-token decision-making, IGR extracts a global routing signal from the user instruction to select a unified "committee of experts" for all spatial locations within the same layer of the model. This globally consistent expert selection strategy is designed to maintain the semantic and structural integrity of the generation process. Furthermore, to enhance functional diversity among experts and prevent "expert collapse," the authors introduce an output-space orthogonality loss. Extensive experiments demonstrate that InstructMoLE significantly outperforms existing LoRA and MoLE variants on multiple challenging multi-conditional generation benchmarks.

**Strengths:**

1.Novel Routing Paradigm: IGR redefines MoLE routing by tying expert selection to global instruction semantics, addressing a fundamental misalignment in prior token-level approaches. The fusion of T5 (compositional) and CLIP (holistic) embeddings via Perceiver attention is a creative combination of existing tools to solve a new problem.
2.Targeted Regularization: Orthogonal loss fills a gap in MoE training—unlike load-balancing losses (which only ensure uniform expert usage), it directly enforces distinct expert functions, a unique design for low-rank MoE.
3.Experimental Comprehensiveness: The paper validates performance across diverse tasks (in-context generation, multi-subject synthesis, single-image editing, spatial control) and includes ablation studies for routing policies, IGR components, and hyperparameters—leaving no critical design choice untested.

**Weaknesses:**

1.Extreme Scenario Testing: No experiments on low-quality instructions (e.g., vague descriptions, typos) or ultra-complex instructions (e.g., 6+ subjects with overlapping spatial relationships). These scenarios are common in practice, and their omission limits the paper’s practical relevance.
2. Ambiguity in the Expert Diversity Mechanism: While the orthogonality loss proves effective, the paper does not elucidate whether it functions solely as a regularizer or if it genuinely fosters functional differentiation among the experts. It remains unclear if the observed benefits stem from a general regularization effect or from the successful cultivation of specialized expert modules.
3. Insufficient Analysis of Computational Overhead: The paper states that the output-space orthogonality loss requires computing the outputs of all experts. This implies that for calculating this loss, a forward pass through all N experts must be performed for the entire data batch, even for an MoE layer that only activates top-k experts during a standard forward pass. This approach could introduce significant computational and memory overhead, potentially negating some of the efficiency benefits of the MoE architecture. We recommend that the authors provide a concise analysis in the appendix, discussing the impact on training time and resource consumption and comparing these metrics against a baseline model trained without this loss.

**Questions:**

1. On the Impact of IGR on Stochastic Diversity

Given that the Instruction-Guided Routing (IGR) mechanism selects a deterministic committee of experts based on the input prompt, does this fixed routing strategy impact the stochastic diversity inherent in a single generation instance? In other words, could it potentially reduce the random variations between different outputs generated from the same prompt and seed?

2. Robustness and Dependency on the Global Signal

The model's decision-making process appears to be highly dependent on the single global signal extracted from the text instruction. Does this imply that the quality of routing is entirely contingent upon how accurately and comprehensively this global signal captures the user's intent? Could this create a robustness issue, making the model sensitive to ambiguous, noisy, or poorly formulated instructions?

3. Router Behavior with Highly Complex Instructions

When faced with exceedingly complex instructions that involve multiple attributes, is the router forced to make trade-offs? For instance, does it prioritize dominant instructions while neglecting minor details? Alternatively, does it default to selecting a generic, "jack-of-all-trades" committee of experts that performs sub-optimally on all aspects of the prompt?

4. Potential Side Effects of Output-space Orthogonality Loss

Regarding the Output-space Orthogonality Loss: could this constraint inadvertently hinder the model's ability to learn complex capabilities that require synergistic collaboration among multiple experts? By enforcing functional separation, does it risk producing images that, while accurate in individual components, may lack holistic harmony and coherence?

5. Ablation Studies on the Orthogonality Loss

Have ablation studies been conducted specifically on the orthogonality loss? For example, was an experiment run using only the orthogonality loss without the IGR mechanism to isolate its effect? Furthermore, is it possible that simply increasing expert capacity (e.g., adding more experts) could serve as an alternative to achieve expert diversity, thereby replacing the need for this loss?

---

> ### Author Response · Authors · 2025-11-21
>
> # (1/8) Extreme Scenario Testing
> We thank the reviewer for emphasizing practical robustness. We have conducted a comprehensive stress test on low-quality and ultra-complex instructions, with qualitative results included in **Appendix A.5 (Figure 12)**.
>
> We evaluated 5 extreme cases ranging from severe typos to logical inference. The results show that InstructMoLE significantly outperforms the baseline in **implicit reasoning** and **abstract interpretation**, while sharing certain foundation-level limitations in extreme noise or load.
>
> 1.  **Success: Robustness in Reasoning and Abstraction**
>     * **Implicit Logic:** In Case #4 ("The person is cold"), our model correctly inferred the causal need to generate clothing (a jacket), a logical leap the baseline failed to execute.
>     * **Abstract Atmosphere:** In Case #2 ("Make it look dangerous"), InstructMoLE successfully altered lighting and expression to create a tense atmosphere, whereas the baseline output remained incongruously bright.
>
> 2.  **Limitations: Analysis of Failure Modes**
>     * **Severe Typos (Case #1):** Both models failed to resolve instructions with heavy textual corruption. This indicates that the CLIP anchor cannot fully compensate when T5 embeddings are severely distorted.
>     * **Ultra-Complex Scenes (Case #5):** In scenarios involving 4 distinct subjects with multiple reference identities, both models struggled with correct **identity binding**.
>         * *Cause (Text-Visual Misalignment):* Our analysis attributes this to the ambiguity of generic textual descriptors (e.g., simply "boy" or "girl") relative to the specific visual references. This vagueness prevented the correct mapping between textual tokens and visual attributes. Consequently, the generation process became **text-dominated**: the strong semantic priors associated with the text tokens overshadowed the specific visual constraints from the reference images, leading to the loss of fine-grained reference identity details.

---

> ### Author Response · Authors · 2025-11-21
>
> # (2/8) Ambiguity in the Expert Diversity Mechanism
>
> We clarify that $L_\text{ortho}$  functions fundamentally differently from standard regularization (e.g., Weight Decay). While standard regularizers constrain model **complexity** (magnitude), $L_\text{ortho}$ enforces **functional independence** (directionality/rank).
>
> We provide both a formal proof and empirical evidence to demonstrate that the observed benefits stem from **genuine functional differentiation**.
>
> ## **1. Proof: $L_{\text{ortho}}$ Enforces Full-Rank Expert Diversity**
>
> Established theory proves that **Expert Diversification** significantly enhances learning performance by mitigating task interference[1]. We prove *how* our loss function mathematically guarantees this diversification by forcing the experts to form a full-rank basis.
>
> **Proposition 1: Minimizing $L_{\text{ortho}}$ Maximizes Functional Rank**
> Let $U(X)=[u_1(X),\dots,u_N(X)]$ be the matrix of normalized expert outputs. We define our loss as:
>
> $$
> L_{\text{ortho}} = E_{X}\left[\frac{1}{N(N-1)}\sum_{i\neq j}(u_i^T u_j)^2\right]
> $$
>
>
> **Proof:**
> Since each column $u_i$ is unit-norm, this loss is mathematically equivalent to the squared Frobenius norm distance between the Gram matrix $U^T U$ and the Identity matrix $I_N$:
> $$
> L_\text{ortho} \equiv \mathbb{E}_X \left[\frac{1}{N(N-1)}\|U^\top U-I_N\|_F^2\right]
> $$
> 1.  **Zero Characterization:** If $\mathcal{L}_{\mathrm{ortho}} \to 0$, then $\|U^\top U - I_N\|_F^2 \to 0$, implying $U^\top U \to I_N$. This means the expert outputs become pairwise orthogonal.
> 2.  **Full Rank Basis:** Since $I_N$ is full rank, minimizing the loss forces $U(X)$ to have **Rank $N$**. This mathematically excludes "Expert Collapse" (where Rank < $N$) and guarantees that the experts form a **stable, linearly independent functional basis**.
>
> **Conclusion:** This full-rank basis ensures that the expert council $e_{council} = \sum w_j e_j$ retains maximum expressivity to handle complex, multi-objective instructions (e.g., "red apple, green pear").
>
> ## **2. Empirical Verification: Structural Improvements Confirm Specialization**
>
> If $L_\text{ortho}$ were merely acting as a general regularizer, we would expect uniform marginal gains (or degradation) across all tasks. Instead, **Table 5** reveals a distinct pattern consistent with **functional specialization**:
>
> * **Holistic Tasks (Single-Subject):** Performance is largely unchanged (6.12 vs. 6.15).
> * **Structural Tasks (Pose/Canny):** Performance explodes (**Pose F1: +3.2%**, **Canny F1: +2.6%**).
>
> **Interpretation:** Structural tasks require decomposing complex geometry into distinct primitives (e.g., vertical lines vs. horizontal lines). The massive gain here confirms that the experts have differentiated into a **basis of geometric primitives** capable of reconstructing complex structures—a capability that a set of redundant (even if regularized) experts could not achieve.
>
> > [1] Li, Hongbo, et al. (2025). "Theory on mixture-of-experts in continual learning." ICLR.

---

> ### Author Response · Authors · 2025-11-21
>
> # (3/8) Computational Overhead Analysis
>
>
> We conducted a rigorous benchmark on 8 × NVIDIA H100 GPUs comparing our **InstructMoLE** (default configuration $N=8, k=4$) against the standard **Token-based MoLE baseline**.
>
> The results, summarized in **Table A**, demonstrate that InstructMoLE achieves state-of-the-art performance with **lower peak memory usage** than the baseline and **zero inference overhead**, justifying the training computational cost.
>
> **Table A: Comprehensive Comparison of Efficiency & Performance ($N=8, k=4$)**
>
> | Configuration | **Peak Memory** (GB) | **Inference (s/image) batch size=1** | **Performance** (Canny / Pose F1) | **Training (s/step) batch size=4×8)** |
> | :--- | :---: | :---: | :---: | :---: |
> | **Token-based Baseline** | 51.88 | 21.99s | 28.06% / 19.50% | 4.37s |
> | **Ours (w/o Ortho Loss)** | **47.95** | **21.81s** | 38.92% / 37.77% | 4.36s |
> | **Ours (w/ Ortho Loss)** | **51.25** | **21.81s** | **41.51% / 40.97%** | 5.81s |
>
> *Performance metrics sourced from Table 4 and Table 5 of the main paper.*
>
> 1.  **Architectural Efficiency (Lower Memory Footprint):**
>     Contrary to concerns about resource consumption, **InstructMoLE (Full) requires less peak memory (51.25 GB)** than the Token-based Baseline (51.88 GB).
>     * **Reason:** Standard token-level routing requires memory-intensive `gather` and `scatter` operations to handle non-contiguous token assignments. Our Instruction-Guided Routing (IGR) eliminates these operations. This architectural efficiency effectively offsets the memory cost of computing all experts for the orthogonality loss.
>
> 2.  **Zero Inference Overhead:**
>     The orthogonal loss is strictly a **training-only objective**. During inference, the model reverts to a standard sparse execution path (activating only top-$k$ experts). As shown in Table 1, our inference latency is **identical** to the baseline, ensuring no computational penalty for end-users.
>
> 3.  **Strategic Training Investment:**
>     We explicitly trade a ~33% increase in training latency (4.37s→5.81s) for substantial performance gains (e.g., **Pose F1 doubles from 19.50%→40.97%**). By computing outputs for all experts during training, we enforce **dense gradient flow** across the entire parameter space. This "Active Stabilization" prevents the expert collapse common in sparse training and ensures robust functional specialization.
>
> We will add more detailed analysis to Appendix to provide a transparent assessment of the computational trade-offs.
>
> ----
>
> # (4/8) On the Impact of IGR on Stochastic Diversity
>
> **InstructMoLE strictly preserves the intrinsic generative diversity of the backbone model, as diversity is mathematically derived from the initial noise sampling rather than routing stochasticity.** We substantiate this from two perspectives:
>
> 1.  **Theoretical Preservation:**
>     The backbone model (Flux Kontext) is based on **Flow Matching** [1]. In this framework, generation is formulated as an Ordinary Differential Equation (ODE) trajectory mapping a stochastic prior $z_1 \sim \mathcal{N}(0, \mathbf{I})$ to data $z_0$.
>     * **Source of Diversity:** The diversity of the generated output is strictly determined by the sampling of the **initial latent $z_1$**.
>     * **IGR Role:** For any specific instruction, IGR computes a deterministic routing vector, effectively **instantiating a fixed-weight adapter** for that instance. This preserves the deterministic ODE mapping $\Phi_\theta(z_1)$ defined by the backbone. Consequently, varying the random seed $z_1$ traverses different flow trajectories, producing diverse layouts and compositions. This ensures the source of diversity remains decoupled from the routing mechanism.
>
> 2.  **Empirical Evidence (Eliminating Harmful Jitter):**
>     Our experiments demonstrate that stochasticity in the routing layer manifests as **functional disorder** rather than meaningful creative diversity. As shown in **Table 4**, when routing is coupled to local noise (e.g., **Expert Race**), the model suffers from structural collapse (Pose F1: **0.00%**), confirming that weight-level variance disrupts functional consistency. By stabilizing the routing, IGR achieves high structural fidelity (Pose F1: **40.97%**) while retaining the valid content diversity driven by the flow matching process.
>
> > [1] Esser, Patrick, et al. "Scaling rectified flow transformers for high-resolution image synthesis." ICML. 2024.

---

> ### Author Response · Authors · 2025-11-21
>
> # (5/8)  Robustness and Dependency on the Global Signal
>
> The robustness is guaranteed by the stability of the fused signal and the intrinsic capabilities of the pre-trained backbone.
>
> ## **1. Signal Stability via Fusion:**
> The global signal ($Z_{global}$) is inherently stable because it is not a fragile single embedding, but a fusion of **CLIP** (which serves as a robust semantic anchor) and **T5** (which provides detailed compositional information). This hybrid design ensures that the routing mechanism receives comprehensive and stable guidance for selecting the appropriate expert committee, minimizing sensitivity to local noise in the prompt.
>
> ## **2. Foundation Model Capabilities:**
> InstructMoLE operates as a parameter-efficient fine-tuning framework (LoRA). The underlying pre-trained MM-DiT backbone retains its intrinsic capability to process and attend to T5 instruction tokens via standard cross-attention mechanisms. Therefore, the model's responsiveness to user intent is doubly secured: the **Global Signal** directs the adaptation (expert selection), while the **Pre-trained Backbone** handles the generative execution. This dual-path processing ensures robust and correct responses to user instructions.
>
> ----
>
> # (6/8) Router Behavior with Highly Complex Instructions
>
> **The reliance on the global signal ensures stable and faithful execution of user intent rather than creating a robustness issue.** This is supported by the architectural stability of our signal design and validated by generalization performance on complex benchmarks.
>
> ## **1. Signal Stability (Ablation Analysis):**
> The global signal is anchored by the **CLIP pooled embedding**, which is natively utilized by the pre-trained Flux.1 kontext backbone for normalization, ensuring inherent stability. We augment this with **T5 details** to enhance precision without compromising this stability.
>     * **Evidence:** As shown in **Table B**, the baseline model using only the CLIP signal already achieves stable performance. However, introducing our fused signal ($Z_{global}$) significantly boosts structural alignment metrics (Pose F1: **36.74% $\to$ 37.77%**) and Multi-Subject fidelity (**64.54 $\to$ 65.65**). This confirms that the added signal provides critical detail utility while maintaining the robustness of the semantic anchor.
>
> **Table B: Ablation study of the IGR signal.**
> | IGR Signal | Multi-Subject (↑) | Depth MSE (↓) | Canny F1 (↑) | Pose F1 (↑) |
> | :--- | :---: | :---: | :---: | :---: |
> | $CLIP(I_c)$ | 64.54 | 35.67 | 38.12 | 36.74 |
> | $Z_{global}$ | **65.65** | **33.74** | **38.92** | **37.77** |
>
> ## **2. Robustness to Interference (Complex-Edit Benchmark):**
> To prove that the global signal is not susceptible to interference from noise or ambiguity, we tested the model on the **Complex-Edit benchmark**, which features highly complex, chain-of-thought instructions.
>
> Despite not being fine-tuned on this specific dataset (training data detailed in Figure 6), InstructMoLE demonstrates robust generalization, outperforming the strong Flux.1 Kontext baseline.
>
> **Table C: Performance on Complex-Edit Benchmark**
>
> | Metric                          | Flux.1 Kontext | InstructMoLE (Ours) |
> |---------------------------------|:--------------:|:-------------------:|
> | **Overall Perceptual Quality (↑)** |     7.08      |       **7.15**      |
> | **Identity Preservation (↑)**      |     8.01      |       **8.09**      |
>
> These results confirm that the dependency on the global signal does not make the model fragile; instead, it enables the router to maintain high identity fidelity (8.09 vs 8.01) even when processing exceedingly complex and potentially noisy instructions.

---

> ### Author Response · Authors · 2025-11-21
>
> # (7/8) Potential Side Effects of Output-space Orthogonality Loss
>
> **Concern:** Does enforcing functional orthogonality hinder synergistic collaboration among experts, risking a lack of holistic harmony in the generated images?
>
> **Response: No.** We argue that functional orthogonality is theoretically grounded and practically essential for maximizing the model's collaborative capacity and holistic coherence.
>
> 1.  **Theoretical Foundation: Diversity Enables Synergy**
>     * **Theoretical Motivation:** Our design is grounded in established theory, which proves that **Expert Diversification** significantly enhances learning performance by mitigating task interference[1]. The Orthogonality Loss is the practical mechanism to enforce this diversity.
>     * **Collaboration via Basis Vectors:** Far from hindering collaboration, orthogonality maximizes it. In a linear combination system like MoLE ($Y = \sum w_i E_i(x)$), synergy is limited if experts are redundant (collinear). By forcing experts to be orthogonal, we ensure they act as distinct "basis vectors" that span a larger functional space. This allows the router to synthesize complex, harmonious behaviors by cleanly combining these non-interfering capabilities.
>
> 2.  **Empirical Evidence of Holistic Harmony**
>     * **Quantitative Validation:** If collaboration were compromised, the model would struggle to integrate multiple distinct elements coherently. However, in the challenging **Multi-Subject Driven Generation** task (Table 2), which demands high synergistic coordination to preserve multiple identities and objects simultaneously, InstructMoLE achieves a superior **Average Score of 71.07**, significantly outperforming the strong Flux.1 Kontext baseline (67.46). This substantial improvement (+3.6) confirms that functional disentanglement enhances, rather than hinders, the model's capacity to harmoniously compose complex scenes.
>     * **Qualitative Validation:** As shown in **Figure 2**, InstructMoLE excels in **Relighting** tasks. Successfully integrating a subject into a new lighting environment requires tight, harmonious collaboration between structure and color experts. Our success here demonstrates that functional separation leads to cleaner disentanglement, which in turn permits more precise and cohesive recombination.
>
>
> > [1] Li, Hongbo, et al. (2025). "Theory on mixture-of-experts in continual learning." ICLR.

---

> ### Author Response · Authors · 2025-11-21
>
> # (8/8) Ablation Studies on the Orthogonality Loss
>
> **We conducted ablation studies to isolate the effect of the Orthogonality Loss and to determine if expert capacity can substitute for functional diversity.**
>
> 1.  **Isolation of Orthogonality Loss:**
>     We analyzed the impact of $\mathcal{L}_{ortho}$ independently of the IGR signal ($Z_{global}$). As shown in **Table D**, adding $\mathcal{L}_{ortho}$ to the baseline configuration (`CLIP` signal only) increases **Canny F1 from 38.12% to 40.22%**. When combined with the advanced signal, the loss further improves performance (e.g., **Pose F1 increases to 40.97%**), indicating a distinct contribution to model performance regardless of the routing signal.
>
>     **Table D: Ablation of Orthogonality Loss Independent of Signal**
>
>     | Model | IGR Signal | $\mathcal{L}_{ortho}$ | Multi-Subject (↑) | Pose F1 (↑) | Canny F1 (↑) |
>     | :--- | :--- | :---: | :---: | :---: | :---: |
>     | **Baseline** | $CLIP(I_c)$ | ✗ | 64.54 | 36.74% | 38.12% |
>     | **+ $\mathcal{L}_{ortho}$ Only** |$CLIP(I_c)$ | ✔ | 64.66 | 36.79% | **40.22%** |
>     | **+ $Z_{global}$ Only** | $Z_{global}$ | ✗ | 65.65 | 37.77% | 38.92% |
>     | **Full Model (Ours)** | $Z_{global}$ | ✔ | **65.68** | **40.97%** | **41.51%** |
>
> 2.  **Capacity vs. Diversity Analysis:**
>     We evaluated whether increasing expert capacity serves as an alternative to enforced diversity.
>     * **Diversity vs. Capacity:** We compared configurations with similar total parameter budgets but different expert counts ($N$) and ranks ($r$). As shown in **Table E**, the high-diversity configuration ($N=8, r=32$) yields higher scores than the high-capacity configuration ($N=2, r=128$), specifically in spatial alignment (**Pose F1: 40.97% vs. 34.02%**).
>     * **Sparsity vs. Density:** Increasing the number of active experts (Dense $k=8$) results in lower performance compared to sparse activation ($k=4$), as shown in **Table F**. This indicates that sparsity combined with functional orthogonality is more effective than dense activation.
>
>     **Table E: Impact of Expert Diversity vs. Capacity**
>
>     | Configuration | Expert Count | Expert Rank | Multi-Subject (↑) | Pose F1 (↑) | Canny F1 (↑) |
>     | :--- | :---: | :---: | :---: | :---: | :---: |
>     | Low Diversity | $N=2$ | $r=128$ | 65.10 | 34.02% | 37.52% |
>     | Med Diversity | $N=4$ | $r=64$ | 65.06 | 34.77% | 38.27% |
>     | **High Diversity** | **$N=8$** | **$r=32$** | **65.68** | **40.97%** | **41.51%** |
>
>     **Table F: Impact of Activation Density**
>
>     | Activation ($k$) | 2 | **4 (Ours)** | 6 | 8 (Dense) |
>     | :--- | :---: | :---: | :---: | :---: |
>     | **Canny F1** | 35.10% | **41.51%** | 37.53% | 37.42% |

---

### Official Review · Reviewer_CVZA · 2025-10-28

**Soundness:** 3
**Presentation:** 2
**Contribution:** 3
**Rating:** 4
**Confidence:** 3

**Summary:**

InstructMoLE is a novel framework for instruction-guided fine-tuning of diffusion transformers. It replaces per-token routing in MoLE with global Instruction-Guided Routing (IGR), ensuring coherent expert selection across all tokens to preserve global semantics. An output-space orthogonality loss encourages expert diversity and prevents representational collapse. Experiments show that InstructMoLE outperforms existing LoRA and MoLE approaches on multi-conditional image generation tasks, offering robust and generalizable instruction-driven control.

**Strengths:**

1. Resolves the mismatch between token-level routing and the global semantic scope of instructions through InstructMoLE with Instruction-Guided Routing.
2. Uses an output-space orthogonality regularizer to prevent expert collapse and improve compositional control.
3. Achieves state-of-the-art performance on multi-conditional image generation, showing the superiority of global, instruction-guided routing.

**Weaknesses:**

1. Lack of a deeper theoretical explanation for why global routing outperforms token-level routing.
2. The author could improve the visual quality and aesthetics of the figures.
3. What is the reason that Expert Race completely fails in Table 4 (Pose F1 = 0%)?
4. The details of Perceiver Attention are not clearly specified, and the motivation for using it in this work is unclear.
5. Scalability: How do performance and efficiency change as the number of experts N increases?
6. Efficiency metrics such as total training cost and inference speed are not reported.

**Questions:**

Please refer to the Weaknesses section.

---

> ### Author Response · Authors · 2025-11-21
>
> # (1/6) Theoretical Explanation
> Our theoretical analysis proves the superiority of our framework by establishing two key propositions:
>
> (1) $L_{\text{ortho}}$ mathematically enforces the expert diversity required for high performance.
>
> (2) IGR structurally eliminates the spatial inconsistency inherent to Token-Level routing.
>
> ---
> ## **1. Proof: $L_{\text{ortho}}$ Enforces Full-Rank Expert Diversity**
>
> Established theory proves that **Expert Diversification** significantly enhances learning performance by mitigating task interference[1]. We prove *how* our loss function mathematically guarantees this diversification by forcing the experts to form a full-rank basis.
>
> **Proposition 1: Minimizing $L_{\text{ortho}}$ Maximizes Functional Rank**
> Let $U(X)=[u_1(X),\dots,u_N(X)]$ be the matrix of normalized expert outputs. We define our loss as:
>
> $$
> L_{\text{ortho}} = E_{X}\left[\frac{1}{N(N-1)}\sum_{i\neq j}(u_i^T u_j)^2\right]
> $$
>
>
> **Proof:**
> Since each column $u_i$ is unit-norm, this loss is mathematically equivalent to the squared Frobenius norm distance between the Gram matrix $U^T U$ and the Identity matrix $I_N$:
> $$
> L_{\mathrm{ortho}} \equiv \mathbb{E}_X \left[\frac{1}{N(N-1)}\|U^\top U-I_N\|_F^2\right]
> $$
> 1.  **Zero Characterization:** If $\mathcal{L}_{\mathrm{ortho}} \to 0$, then $\|U^\top U - I_N\|_F^2 \to 0$, implying $U^\top U \to I_N$. This means the expert outputs become pairwise orthogonal.
> 2.  **Full Rank Basis:** Since $I_N$ is full rank, minimizing the loss forces $U(X)$ to have **Rank $N$**. This mathematically excludes "Expert Collapse" (where Rank < $N$) and guarantees that the experts form a **stable, linearly independent functional basis**.
>
> **Conclusion:** This full-rank basis ensures that the expert council $e_{council} = \sum w_j e_j$ retains maximum expressivity to handle complex, multi-objective instructions (e.g., "red apple, green pear").
>
> ---
>
> ## **2. Proof: IGR Eliminates Spatial Inconsistency**
>
> **Spatial inconsistency degrades image quality.** Structure‑ and gradient‑aware image quality assessment objectives explicitly penalize local discontinuities and boundary mismatches; higher seam count/magnitude receives larger penalties and correlates with lower perceptual scores [2, 3]. Consistently, recent harmonization/blending/inpainting/editing methods that suppress seams report improved objective metrics and human preference [4, 5, 6]. Consequently (all else equal), more seams imply a larger loss and reduced perceived quality.
>
> **Proposition 2: IGR Structurally Guarantees Consistency**
>
> **Consistency Condition:** For any two adjacent tokens $x_i$ and $x_j$ belonging to the same semantic object, their routing decisions must be identical: $R(x_i) = R(x_j)$.
>
> **Proof of TL Failure:**
> Token-Level (TL) routing makes decisions from local states: $R_{TL}(x_i) = \arg\max g(x_i)$. Since local states inherently differ ($x_i \neq x_j$) due to noise or feature variations, the independent optimization of $R_{TL}$ allows divergent outcomes under the same instruction $I$:
> $$
> \mathbb{P}\left(R_{TL}(x_i) \neq R_{TL}(x_j) \mid I\right) > 0.
> $$
> This non-zero probability constitutes spatial fragmentation: the model applies different functions to adjacent parts of the same object, violating the consistency condition.
>
> **Proof of IGR Success:**
> Instruction-Guided Routing (IGR) decouples the decision from local states. It selects a single council $e_{council}$ based on $I$ and broadcasts it to all tokens:
> $$
> R_{IGR}(x_i) = R_{IGR}(x_j) = e_{council}, \quad \forall i,j.
> $$
> Therefore,
> $$
> \mathbb{P}\left(R_{IGR}(x_i) \neq R_{IGR}(x_j) \mid I\right) \equiv 0,
> $$
> and spatial consistency holds by design.
>
> Thus, unlike TL routing, IGR satisfies the consistency condition by construction and eliminates seams.
>
> > [1] Li, Hongbo, et al. (2025). "Theory on mixture-of-experts in continual learning." ICLR.
> > [2] Ding, K. et al. (2020). Image Quality Assessment: Unifying Structure and Texture Similarity.
> > [3] Andersson, P.-E. et al. (2020). FLIP: A Difference Evaluator for Image Quality.
> > [4] Cong, W. et al. (2020). DoveNet: Deep Image Harmonization via Domain Verification. CVPR.
> > [5] Suvorov, R. et al. (2022). Resolution-robust Large Mask Inpainting with Fourier Convolutions. WACV.
> > [6] Avrahami, O. et al. (2022). Blended Diffusion for Text-driven Editing of Natural Images. CVPR.

---

> ### Author Response · Authors · 2025-11-21
>
> # (2/6) Improvement of Visual Quality and Layout
>
> We thank the reviewer for this constructive suggestion. We will promptly modify the layout of the tables and figures to provide a more polished and comprehensive version in the revised manuscript.
>
> ----
>
> # (3/6) Analysis of Baseline Failure: Why Expert Race Collapses on Pose Alignment
>
> Expert Race employs an **"unlimited flexibility"** strategy that performs a **global Top-K selection** across all tokens in the batch. While beneficial for general text-to-image tasks, this mechanism is structurally destructive for spatially structured tasks like Pose Estimation.
>
> ## **1. Structural "Dropout" via Global Competition**
> Unlike standard routing which guarantees a fixed number of experts per token, Expert Race forces tokens to **compete against each other** for capacity.
> * **The Failure Mode:** Pose skeletons are sparse, thin structures. If these structural tokens generate slightly lower activation logits than texture-rich background tokens, they "lose the race" and effectively receive **zero expert capacity** (or are filtered out by the dynamic threshold $\tau$ ).
> * **Result:** The model effectively "drops" the pose structure entirely, explaining the **0.00% score**.
>
> ## **2. Maximum Spatial Fragmentation**
> Expert Race treats every token-expert pair as an independent candidate.
> * **The Failure Mode:** This maximizes the probability of routing conflicts between adjacent tokens ($\mathbb{P}(R(x_i) \neq R(x_j)) \approx 1$). For a connected geometric structure like a skeleton (shoulder $\to$ elbow $\to$ wrist), this discontinuity breaks the long-range dependencies required to form a coherent figure.
>
> **Conclusion:**
> Expert Race prioritizes **token-level competition** over **spatial coherence**. In contrast, **IGR** enforces spatial consistency by broadcasting a single, instruction-aware council to all tokens ($R(x_i) \equiv R(x_j)$), preserving the global structural integrity required for pose generation.
>
> ----
>
> # (4/6) Details and Motivation of Perceiver Attention
>
> We employ Perceiver Attention as a **Semantic Information Bottleneck** to resolve a critical architectural conflict: how to compress variable-length, multi-modal conditions into a strictly fixed-length routing signal without losing fine-grained details.
>
> ## **1. Motivation: Resolving the Conflict between Consistency and Adaptability**
>
> **The Conflict:** Our theoretical framework (**Proposition 2**) mandates a **single, fixed-length** global signal ($Z_{global}$) to enforce spatial consistency ($R(x_i) \equiv R(x_j)$). However, our inputs are **variable-length** and heterogeneous (e.g., T5 text tokens + Pose maps + Depth maps).
>
> **The Solution:** Simple pooling (Mean/Classification token) satisfies the "fixed-length" requirement but fails to capture fine-grained compositional details. Perceiver Attention bridges this gap via two mechanisms:
> * **Structural Constraint (for Consistency):** It forces all inputs into a fixed latent shape, mathematically guaranteeing that the router sees only a global context.
> * **Adaptive Weighting (for Adaptability):** Unlike static pooling, its **Latent Cross-Attention** dynamically weighs modalities. It can learn to "attend" to sparse Pose tokens when structural control is needed, or Text tokens for style, ensuring the compressed $Z_{global}$ retains high fidelity.
>
> ## **2. Methodology: Implementation Details**
>
> We configure the Perceiver block to act as a learnable query mechanism:
> * **Fixed Query ($Q$) $\rightarrow$ Enforces Consistency:** We initialize a *single* learnable latent query vector ($Q \in \mathbb{R}^{1 \times 1 \times D}$). By definition, the output of the attention mechanism matches the shape of this query. This physically forces the output to be a single vector, preventing the router from accessing spatially varying features.
> * **Cross-Attention ($K, V$) $\rightarrow$ Enables Adaptability:** We compute $\text{Attention}(Q, K=X_{cond}, V=X_{cond})$. This allows the single query to "scan" the entire variable-length input sequence ($X_{cond}$) and selectively extract the most relevant attribute-binding signals (e.g., separating "red" from "apple") into $Z_{global}$.
> * **Lightweight Structure** It consists of only 2 cross-attention layers with a single query token. Compared to the DiT backbone, it introduces **negligible parameters and computational overhead (～ 0.1% )**, ensuring that the performance gains come from better routing, not increased model capacity.
>
>
> ## **3. Empirical Validation**
>
> Table 5 demonstrates that this architectural choice drives comprehensive performance gains across all tasks. Comparing the **Baseline** (CLIP) routing to our **Perceiver-based routing** ($Z_{global}$ ):  We observe consistent gains across both compositional tasks (**Multi-Subject**: 64.54 $\to$ 65.65) and spatial tasks (**Pose F1**: 36.74% $\to$ 37.77%; **Canny F1**: 38.12% $\to$ 38.92%).

---

> ### Author Response · Authors · 2025-11-21
>
> # (5/6) Performance and Efficiency w.r.t Expert Count ($N$)
>
> We analyze scalability within the context of **Parameter-Efficient Fine-Tuning (PEFT)**. Our experiments specifically investigate how to maximize performance efficiency given a fixed computational budget.
>
> **1. Performance Scalability: Diversity Outperforms Capacity**
> Based on the ablation study in **Table 6 of our paper**, we compared varying expert counts ($N$) while maintaining a constant activation budget ($r \times k \approx 128$) to isolate the benefit of expert diversity from raw parameter increase.
>
> | Configuration | Multi-Subject (↑) | Pose F1 (↑) | Canny F1 (↑) |
> |:---|:---:|:---:|:---:|
> | Low $N$, High Rank ($N=2, r=128$) | 65.10 | 34.02% | 37.52% |
> | Med $N$, Med Rank ($N=4, r=64$) | 65.06 | 34.77% | 38.27% |
> | **High $N$, Low Rank ($N=8, r=32$)** | **65.68** | **40.97%** | **41.51%** |
>
> **Key Insight:** Scaling $N$ (from 2 to 8) yields significantly better returns than scaling expert rank ($r$). Specifically, increasing $N$ boosts Pose F1 by **+6.95%**, validating our theoretical claim: **expert diversity (enforced by $\mathcal{L}_{ortho}$) is the primary driver of scalability in multi-conditional generation**, efficiently covering the diverse functional space required by complex instructions.
>
>
> **2. Interference Scalability: The Critical Role of Sparsity**
> As $N$ increases, managing interference is key. For our fixed pool ($N=8$), performance peaks at sparse activation ($k=4$) rather than dense activation ($k=8$).
>
> | Activation ($k$) | 2 | **4** | 6 | 8 (Dense) |
> |:---|:---:|:---:|:---:|:---:|
> | **Canny F1** | 35.10% | **41.51%** | 37.53% | 37.42% |
>
> **Analysis:** Increasing $N$ requires **sparse activation** to prevent interference. The drop in dense mode ($k=8$) proves that selectivity is essential for our design ($k=4$) successfully disentangles diverse experts.
>
> **3. Efficiency Scalability: Architectural Superiority ($O(N)$ vs $O(N \cdot L)$)**
> InstructMoLE exhibits superior scaling properties compared to standard MoE methods:
> * **Constant Inference FLOPs:** Since we use a fixed activation budget ($Top\text{-}k$), the computational cost of the expert layer remains **constant** regardless of the total pool size $N$.
> * **Routing Efficiency ($O(N)$):** Standard Token-Level routing scales linearly with image resolution ($L$), as it computes routing for every token ($O(N \cdot L)$). In contrast, **Instruction-Guided Routing (IGR)** computes the decision **once per instruction**, scaling only with expert count ($O(N)$). This structural advantage makes InstructMoLE significantly more efficient to scale to massive expert pools or high-resolution generation.

---

> ### Author Response · Authors · 2025-11-21
>
> # (6/6) Efficiency Metrics and Cost Analysis
>  We addressed the lack of efficiency metrics by benchmarking **InstructMoLE** ($N=8, k=4$) against the **Token-based Baseline** on 8 $\times$ NVIDIA H100 GPUs.
>
> **Conclusion:** InstructMoLE achieves state-of-the-art performance with **lower peak memory** and **zero inference overhead**, effectively justifying the moderate increase in training cost.
>
> **Table A: Comprehensive Efficiency & Performance Comparison**
>
> | Configuration | **Peak Memory** (GB) | **Inference Latency** (Batch=1) | **Training Latency** (Batch=32) | **Relative Cost** | **Performance** (Pose F1) |
> | :--- | :---: | :---: | :---: | :---: | :---: |
> | **Token-based Baseline** | 51.88 | 21.99s | 4.37 s/step | 1.0x | 19.50% |
> | **Ours (w/o Ortho Loss)** | **47.95** | **21.81s** | **4.36 s/step** | **1.0x** | 37.77% |
> | **Ours (Full Model)** | **51.25** | **21.81s** | 5.81 s/step | 1.33x | **40.97%** |
>
> **Analysis:**
>
> 1.  **Strategic Training Investment (High ROI):**
>     Training latency increases by ~33% (1.33x cost). This is a necessary investment for stability: dense gradient flow prevents expert collapse. Crucially, this moderate cost yields a **2x improvement in Pose F1 (19.50% $\to$ 40.97%)**, representing a highly favorable trade-off.
>
> 2.  **Architectural Efficiency (Lower Memory):**
>     InstructMoLE (Full) requires **less peak memory (51.25 GB)** than the Baseline (51.88 GB).
>     * *Reason:* Our Instruction-Guided Routing (IGR) eliminates the memory-intensive `gather` and `scatter` operations required by token-level routing, effectively offsetting the cost of computing all experts.
>
> 3.  **Zero Inference Overhead:**
>     The orthogonality loss is a **training-only objective**. During inference, the model reverts to a standard sparse execution path. As shown in Table A, our inference latency is **identical** to the baseline, ensuring no penalty for deployment.
>
> We will include this detailed efficiency breakdown in Appendix.

---

### Official Review · Reviewer_vXq2 · 2025-10-31

**Soundness:** 3
**Presentation:** 3
**Contribution:** 3
**Rating:** 6
**Confidence:** 4

**Summary:**

The paper introduces a novel instruction-guided instance-level routing policy for MoLE in image editing tasks. This approach ensures that all tokens in an image share the same set of routed experts across different network layers, effectively addressing issues such as global inconsistency and semantic. Overall, the paper presents Instruction-Guided Routing (IGR) as a compelling innovation to align MoLE with global instructions, also effectively adopting existing methods such as Perceiver Attention, LoRA, and orthogonality regularization to solve information loss, computational overhead and expert collapse.

**Strengths:**

1. The paper successfully adapts the MoLE framework to DiT-based image editing tasks, specifically addressing the conflict between the global consistency requirements inherent in image editing and the limitations of token-level routing.
2. It employs an output-space orthogonality regularization to mitigate the problem of expert collapse, and efficiently simplifies the associated computational cost by introducing the Gram matrix.
3. The experimental setup is comprehensive, including comparisons across various image editing task types and their corresponding standard metrics, complemented by thorough ablation studies.

**Weaknesses:**

1. Table 4 shows that the token-level Expert Choice (EC) method slightly outperforms IGR in the Multi-Subject task. This result is somewhat counter-intuitive, as the Multi-Subject task inherently demands the global consistency understanding that IGR is designed to provide.
2. Some evaluation metrics rely heavily on GPT-4.1 assessments, which may introduce biases and inconsistencies. While this is common in the field, more diverse evaluation approaches or human studies would strengthen the validation.
3. The paper focuses primarily on successful cases but provides limited analysis of scenarios where the proposed approach might fail or perform poorly.

**Questions:**

1. What is the computational overhead of the Perceiver-style attention mechanism compared to standard token-level routing?
2. How does performance change with different numbers of experts (N) and activation patterns (k)? Is there an optimal ratio, and how does this relate to task complexity?

---

> ### Author Response · Authors · 2025-11-21
>
> # (1/5) Analysis of Expert Choice (EC) Performance vs. IGR in Multi-Subject Tasks
>
> The slight advantage of Expert Choice (EC) on the Multi-Subject task (65.81 vs. 65.68) is not a failure of IGR, but a predictable result of the trade-off between **Local Feature Fidelity** and **Global Structural Consistency**.
>
> **1. Mechanism Comparison: Independent Selection vs. Global Broadcast**
>
> The two methods fundamentally differ in how they allocate compute resources across the image lattice.
>
> | Feature | Expert Choice (EC) | InstructMoLE (IGR) |
> | :--- | :--- | :--- |
> | **Routing Logic** | **Expert-Centric (Inverted)** | **Instruction-Centric (Global)** |
> | **Mechanism** | Experts independently select the Top-$k$ tokens they are best suited for from the image sequence. | The router selects a single Expert Council based on the instruction, **broadcasting** it to *all* tokens. |
> | **Spatial Behavior** | **Heterogeneous:** Token $x_i$ and $x_j$ are processed by different subsets of experts based on local features. | **Homogeneous:** Token $x_i$ and $x_j$ are processed by the **identical** expert combination ($e_{council}$). |
> | **Advantage** | **Local Fidelity:** Experts can specialize (e.g., "Fur Expert" processes only animal pixels), maximizing texture quality. | **Spatial Consistency:** Structurally guarantees that adjacent tokens undergo the same transformation. |
> | **Trade-off** | **Spatial Fragmentation:** Breaks geometric dependencies as adjacent tokens follow disjoint compute paths. | **Capacity Sharing:** The council must handle all subjects simultaneously, slightly diluting focus on specific textures. |
>
> **2. Analysis of Performance Differences**
>
> * **Why EC Wins on Multi-Subject (+0.13):**
>     The Multi-Subject metric primarily assesses **object identity and texture**. EC's mechanism allows specific experts (e.g., a "Dog Expert") to selectively process only the relevant tokens (the dog) while ignoring the rest. This "divide-and-conquer" approach maximizes the purity of local features, yielding a marginal gain in texture fidelity.
>
> * **Why EC Fails on Spatial Tasks (-9.50% Pose):**
>     EC's independent selection creates **invisible seams**. If a token on a "shoulder" is selected by *Expert A* and an adjacent token on the "arm" is selected by *Expert B*, the geometric continuity is severed. This **spatial fragmentation** renders EC incapable of maintaining the global structural integrity required for Pose Estimation (31.47% vs. IGR's 40.97%).
>
> * **Why IGR is the Superior Global Solution:**
>     IGR prioritizes **coherence**. By enforcing a single, consistent function across the entire image, it mathematically eliminates spatial fragmentation. While this requires the expert council to share capacity between multiple subjects (causing the negligible 0.13 drop), it secures the **structural stability** essential for complex generative tasks, as evidenced by good performance on Pose and Canny benchmarks.

---

> ### Author Response · Authors · 2025-11-21
>
> # (2/5) Evaluation Metrics
>
> We sincerely appreciate the reviewer’s valuable feedback regarding evaluation methodologies. We acknowledge the valid concern that **the specific metrics relying on GPT-4.1 assessments** (such as those for the GEdit benchmark) may introduce biases. We fully recognize the importance of ensuring robustness through diverse evaluation approaches.
>
> ## **1. Human Evaluation Study**
> We conducted a blind human preference study to rigorously assess model performance beyond automated metrics.
> * **Methodology:** We randomly sampled **30 comparison groups** from the test sets of the three benchmarks reported in the paper (Tables 1, 2, and 3). For each group, the outputs of different models were randomly arranged to ensure a blind evaluation. Human evaluators were asked to perform **multiple-choice selection** (allowing ties) based on two criteria: **Instruction Following** and **Image Quality**.
> * **Results:** As shown in the table below, **InstructMoLE** consistently achieved the highest selection frequency in both categories, demonstrating superior performance aligned with human perception.
>
> | Model | Instruction Following (Selections) | Image Quality (Selections) |
> | :--- | :---: | :---: |
> | **InstructMoLE (Ours)** | **152** | **131** |
> | Flux.1 Kontext | 150 | 122 |
> | OmniGen2 | 116 | 83 |
> | DreamO | 40 | 28 |
>
> ## **2. Robustness via Multi-Dimensional Objective Metrics**
> We wish to highlight that our current evaluation suite is already designed to minimize reliance on any single metric type. We balance the GPT-based scores required by certain benchmarks with a rigorous set of **objective, deterministic metrics** to verify our core claims.
>
> The table below details the specific metrics used in each experiment, demonstrating that our key contributions (spatial consistency and identity preservation) are primarily validated by objective measures:
>
> | Table in Paper | Metric | Type | Evaluation Purpose |
> | :--- | :--- | :--- | :--- |
> | **Table 1** (OmniContext) | **ID-Sim** (ArcFace) | Objective | Quantitatively measures identity preservation accuracy. |
> | **Table 2** (XVerseBench) | **ID-Sim** (Face ID) | Objective | Measures human identity fidelity. |
> | | **IP-Sim** (DINOv2) | Objective | Measures visual fidelity of objects and subjects. |
> | | **DPG Score** | VLM-based | Measures instruction adherence. |
> | **Table 3** (GEdit) | **GEdit Score** | GPT-based | Evaluates editing quality across 11 categories (Benchmark Standard). |
> | **Table 4 & 5** (Ablations) | **Depth MSE** | Mathematical | Pixel-level measurement of depth alignment accuracy. |
> | | **Canny F1** | Mathematical | Structural edge alignment accuracy. |
> | | **Pose F1** (OKS) | Mathematical | Human pose skeleton alignment based on keypoints. |

---

> ### Author Response · Authors · 2025-11-21
>
> # (3/5) Extreme Scenario Testing and Failure Analysis
>
> We thank the reviewer for highlighting the need for a balanced analysis. To address this, we conducted some tests on low-quality and ultra-complex instructions, documenting both successes and **failure modes** in **Appendix A.5 (Figure 12)**.
>
> Our analysis identifies specific boundaries of the proposed approach:
>
> **Identified Failure Modes:**
> * **Severe Textual Corruption (Case #1):** When instructions contain heavy typos (e.g., "Chnage the bckgrnd..."), the model fails to resolve the correct semantics. This indicates a robustness threshold: while our CLIP anchor provides stability, it cannot fully compensate when T5 token embeddings are severely distorted.
>
> * **Ultra-Complex Multi-Reference Binding (Case #5):** In scenarios involving 4 distinct subjects with multiple reference identities, the model struggled with correct **identity binding**.
>     * *Cause Analysis (Text-Visual Misalignment):* We attribute this to the ambiguity of generic text descriptors (e.g., "boy", "girl") relative to specific visual references. This vagueness prevented correct mapping, causing the generation to become **text-dominated**: strong semantic priors from the text tokens overshadowed the specific visual constraints from the reference images.
>
> While InstructMoLE excels in semantic reasoning, robustness to extreme input noise and massive multi-reference binding remain open challenges governed by the modality alignment of the foundation model.
>
> ----
>
>
> # (4/5) Computational Analysis: Efficiency of Perceiver-based IGR
>
> The Perceiver-based IGR mechanism offers a superior efficiency profile compared to standard token-level routing, particularly at high resolutions.
>
> **1. Decoupled Routing Complexity ($O(1)$ vs $O(L)$)**
> * **Token-Level Bottleneck:** Standard MoE routing scales linearly with sequence length ($L$), requiring a routing operation for *every* token (e.g., $O(L \times N)$). This becomes a computational bottleneck at high resolutions.
> * **IGR Advantage:** By compressing the sequence into a single global vector *before* routing, IGR reduces the routing operation to constant complexity ($O(1 \times N)$). The marginal cost of the Perceiver attention (< 0.1% FLOPs) is **structurally offset** by eliminating these expensive per-token routing computations, ensuring high inference throughput.
>
> **2. Negligible Parameter Overhead**
> * **Minimal Footprint:** The Perceiver module adds only ~17.9M parameters. Relative to the 12B parameter FLUX.1 backbone, this represents a mere **0.15% increase** in model size. This confirms that our performance gains stem from superior routing logic, not increased model capacity.

---

> ### Author Response · Authors · 2025-11-21
>
> # (5/5) Scalability and Optimal Configuration
>
> We analyze scalability within the context of **Parameter-Efficient Fine-Tuning (PEFT)**. Based on the ablation study in **Table 6**, we investigate performance scaling across two distinct dimensions: (1) the impact of expert diversity under a fixed computational budget, and (2) the impact of activation sparsity within a fixed expert pool.
>
> ## **1. Scalability of Expert Diversity (Fixed Activation Budget)**
>
> To isolate the benefit of architectural diversity from raw parameter capacity, we scale the number of experts ($N$) while adjusting rank ($r$) and activation ($k$) to maintain a **constant inference cost** ($r \times k \approx 128$).
>
> | Configuration ($N, r, k$) | **Structure** vs. **Capacity** | **Multi-Subject** (↑) | **Pose F1** (↑) | **Canny F1** (↑) |
> | :--- | :--- | :---: | :---: | :---: |
> | $N=2, r=128, k=1$ | Low Diversity, High Capacity | 65.10 | 34.02% | 37.52% |
> | $N=4, r=64, k=2$ | Medium Diversity | 65.06 | 34.77% | 38.27% |
> | **$N=8, r=32, k=4$** | **High Diversity, Low Rank** | **65.68** | **40.97%** | **41.51%** |
>
> **Analysis:** The data reveals a non-linear gain from diversity. While $N=2$ and $N=4$ show comparable performance, increasing to $N=8$ yields a distinct **+6.2% jump in Pose F1**. This suggests a **"diversity threshold"**: a sufficiently large basis of experts ($N \ge 8$) is required to effectively disentangle the complex geometric constraints of structural tasks.
>
> ## **2. Optimality of Activation Sparsity (Fixed Pool $N=8$)**
>
> We fix the expert pool ($N=8, r=32$) and vary the number of activated experts ($k$) to determine the optimal activation strategy.
>
> | Activation ($k$) | Sparsity Regime | **Multi-Subject** (↑) | **Pose F1** (↑) | **Canny F1** (↑) |
> | :---: | :---: | :---: | :---: | :---: |
> | 2 | Under-Activated | 64.60 | 36.47% | 35.10% |
> | **4** | **Optimal Sparsity** | **65.68** | **40.97%** | **41.51%** |
> | 6 | Over-Activated | 64.11 | 37.90% | 37.53% |
> | 8 | Dense (Non-Sparse) | 65.22 | 34.61% | 37.42% |
>
> **Analysis:** Performance peaks at **$k=4$**, confirming the necessity of **sparse activation**.
> * **Why "Dense" degrades Structural Performance:** At $k=8$, the model degenerates into a dense network where all experts process every token. While the high parameter count maintains reasonable semantic performance (Multi-Subject: 65.22), the **loss of sparsity** prevents experts from specializing.
> * **Signal Averaging:** This lack of specialization is revealing in the **Pose F1** score, which drops significantly to **34.61%**. Without distinct, specialized experts to handle conflicting geometric constraints, the model suffers from **signal averaging**, blurring the precise structural control signals.
>
> ## **3. Efficiency Scalability**
>
> * **Inference Latency:** Since we utilize a fixed activation budget ($k \times r$), the FLOPs for the expert layers remain **constant** regardless of the total pool size $N$.
> * **Routing Overhead:** A key advantage of IGR is that routing scales as **$O(N)$** (once per image), whereas token-level routing scales as **$O(N \times S)$** (once per token). This makes our architecture significantly more efficient for high-resolution scaling.

---

> > ### Comment · Reviewer_vXq2 · 2025-11-26
> >
> > Thanks for your detailed response. I'll maintain my rating.

---

### Official Review · Reviewer_51YZ · 2025-11-01

**Soundness:** 3
**Presentation:** 2
**Contribution:** 2
**Rating:** 4
**Confidence:** 3

**Summary:**

This paper introduces InstructMoLE, a Mixture of Low-Rank Experts (MoLE) framework with Instruction-Guided Routing (IGR) for multi-conditional image generation. It addresses token-level routing issues by leveraging global semantics from user instructions and introduces an orthogonality loss to ensure expert diversity. Extensive experiments show state-of-the-art performance across diverse benchmarks.

**Strengths:**

* Clear motivation: The paper clearly identifies the limitations of existing token-level MoE approaches and proposes an instance-level method.

* Technical novelty: The introduction of orthogonal regularization specifically addresses the expert collapse problem in MoE models.

**Weaknesses:**

1. Missing standard benchmarks: The evaluation did not include some widely-used T2I benchmarks such as GenEval and T2I CompBench. Including results on these benchmarks would make the comparison more comprehensive and convincing.

2. Ablation study clarity: Although Table 5 presents ablation results, the signal is not very clear. The current format may give the impression that each row is an independent setting, but the description in the main text suggests that each row accumulates on the previous one. This should be made explicit in the table (e.g., using "+" notation or clearer legends).

  Additionally, from the second row, the improvement from adding Z_global seems marginal, and in some tasks (e.g., single object), performance even drops. Could the authors analyze and explain why this happens?
Furthermore, what are the effects of using only Z_global or only the orthogonal loss? Including these settings would give a more complete view of each component’s contribution.

3. Analysis of orthogonal loss: The application of orthogonal loss is interesting and promising. However, it would be more convincing if the authors could provide a direct comparison of expert weighting distributions with and without orthogonal loss. Such visualization or quantitative analysis would better demonstrate the effectiveness of orthogonal regularization in mitigating expert collapse.

**Questions:**

see weakness

---

> ### Author Response · Authors · 2025-11-21
>
> # (1/4) Evaluation on Standard T2I Benchmarks (GenEval)
>
> We evaluated our model on **GenEval** following the official protocol (553 prompts × 4 seeds = 2,212 images).
>
> **Note:** InstructMoLE is an **editing-specialized model**, fine-tuned *exclusively* on the editing instructions detailed in the Appendix. We performed **zero T2I-specific fine-tuning**.
>
> Despite being trained solely for editing, our model **surpasses the strong FLUX.1-Kontext-dev baseline** on the Overall Score and demonstrates superior compositionality. This confirms that our MoLE architecture not only masters editing instructions but also preserves and even enhances the base model's generative capabilities.
>
> **Table A: GenEval Results (Text-to-Image Benchmark)**
> |Model|Overall|TwoObject|ColorAttr|Position|SingleObject|Counting|Colors|
> |:---|:---:|:---:|:---:|:---:|:---:|:---:|:---:|
> |FLUX.1‑Kontext‑dev|63.57|77.53|50.25|**16.25**|96.56|63.44|77.39|
> |**Ours**|**64.14**|**79.29**|**52.75**|14.00|96.56|**63.75**|**78.46**|
>
> ----
>
> # (2/4) Ablation Study Clarification
>
> We thank the reviewer for this valuable suggestion regarding clarity.
>
> We revised **Table 5** to explicitly present independent configurations, using "+" to denote components added relative to the baseline. We will update Table 5 in the final revision of the paper to explicitly reflect this structure.
>
> **Table 5 (Revised):** Ablation study of the IGR signal and the orthogonality loss. Each row is an independent configuration. (↑ larger is better; ↓ smaller is better.)
>
> | Model | IGR Signal | $\mathcal{L}_{ortho}$ | Multi-Subject (↑) | Single-Subject (↑) | Depth MSE (↓) | Canny F1 (↑) | Pose F1 (↑) |
> | :--- | :--- | :--- | :---: | :---: | :---: | :---: | :---: |
> | Baseline | $CLIP(I_c)$| ✗ | 64.54 | 6.12 | 35.67 | 38.12 | 36.74 |
> | +$\mathcal{L}_{ortho}$ only | $CLIP(I_c)$ | ✔ | 64.66 | 6.12 | 34.55 | 40.22 | 36.79 |
> | +$Z_{global}$ only | $Z_{global}$ | ✗ | 65.65 | 5.97 | 33.74 | 38.92 | 37.77 |
> | **Full Model (Ours)** | $Z_{global}$ | ✔ | **65.68** | **6.15** | **33.34** | **41.51** | **40.97** |
>
> ### **Why does $Z_{global}$ alone hurt Single-Subject performance? (6.12→5.97)**
> * **Native Stability of CLIP:** The CLIP pooled embedding is natively used by the pre-trained Flux kontext backbone for normalization. This signal is inherently stable and sufficient for simple, single-subject generation, even if its semantics are relatively coarse.
> * **Mismatch with Redundant Experts:** $Z_{global}$ fuses T5 features to introduce fine-grained compositional details. However, without $\mathcal{L}_{ortho}$, the expert pool suffers from representation collapse (redundancy). Sending a high-variance, detailed signal to a set of functionally identical ("collapsed") experts introduces routing variance without any functional benefit. This disturbs the inherent stability of the native CLIP signal, causing a slight degradation in simple tasks.
>
> ### **Analysis of Component Contributions:**
>
> 1. **Role of $\mathcal{L}_{ortho}$ (Enforcing Full-Rank Basis):**
> * Theoretically, minimizing this loss forces the expert outputs to be orthogonal, guaranteeing a **Full-Rank Functional Basis** (Proposition 1) . This eliminates "expert collapse" (redundancy), ensuring the model has linearly independent functional primitives to execute spatial tasks effectively.
> 2. **Role of $Z_{global}$ (Providing Precise Guidance):**
> * $Z_{global}$ extracts fine-grained compositional details. However, without $\mathcal{L}_{ortho}$, the expert basis remains **rank-deficient** (redundant). Sending high-variance, detailed instructions to a redundant expert pool introduces routing noise without gaining actual functional expressivity.
> 3. **Synergy (Full Model):**
> * This configuration combines **Full-Rank Diversity** (Prop 1) with **Spatial Consistency** (Prop 2). The precise signal ($Z_{global}$) finally maps to a diverse set of experts, and IGR ensures this optimized "expert council" is applied identically across all tokens, preventing the spatial fragmentation inherent to token-level baselines.
>
> **We provide a detailed theoretical analysis of these conclusions in the end for reference.**

---

> ### Author Response · Authors · 2025-11-21
>
> # (3/4) Analysis of Orthogonal Loss
>
> We appreciate the suggestion. To demonstrate the effectiveness of the orthogonal loss, we provide both the requested quantitative comparison using 1,190 real editing instructions from GEdit benchmark.
>
> Our analysis reveals that $\mathcal{L}_{ortho}$ prevents **Representation Collapse** without disrupting **Routing Balance**, directly translating into performance gains.
>
> 1.  **Routing Balance:**
>     * **Mechanism:** Routing balance is primarily driven by the auxiliary load-balancing loss ($\mathcal{L}_{aux}$).
>     * **Analysis:** As shown below, the baseline (w/o Ortho) already exhibits high routing entropy ($\approx 0.94$). Adding $\mathcal{L}_{ortho}$ provides a modest improvement (+2.4% Entropy). This proves that **enforcing orthogonality does not disrupt the router's ability to distribute load evenly.**
>
>     | Metric | Without Ortho | With Ortho | Change | Interpretation |
>     | :--- | :--- | :--- | :--- | :--- |
>     | **Norm. Entropy** | 0.9380 | **0.9604** | +2.4% | Higher = More balanced |
>     | **Coeff. of Variation** | 0.4743 | **0.3752** | -20.9% | Lower = More balanced |
>
> 2.  **Functional Diversity (The Critical Impact):**
>     * **Mechanism:** While routing is balanced, established theory[1] suggests that experts must be *functionally distinct* to mitigate interference.
>     * **Analysis:** The baseline suffers from severe **Representation Collapse** (Max Redundancy = 0.960), implying that distinct experts are producing nearly identical outputs (routing to "clones"). Adding $\mathcal{L}_{ortho}$ drastically reduces this redundancy (-50.4%), ensuring that balanced routing translates into genuine functional diversity.
>
>     | Metric | Without Ortho | With Ortho | Improvement | Interpretation |
>     | :--- | :--- | :--- | :--- | :--- |
>     | **Max Redundancy** | 0.960 | **0.477** | **-50.4%** | Lower = Less redundant |
>     | **Frobenius Gap** | 14.099 | **4.734** | **-66.4%** | Lower = More orthogonal |
>
> 3.  **Performance Validation (Table 5 Revised):**
>     Finally, we verify that this increased functional diversity translates to tangible performance gains. As shown in our ablation study (**Table 5**), adding *only* $\mathcal{L}_{ortho}$ to the baseline improves **Canny F1 from 38.12% to 40.22%**. This confirms that the loss is not merely a theoretical constraint but a practical driver of model capability.
>
> The quantitative metrics confirm that $\mathcal{L}_{ortho}$ effectively converts "nominal capacity" into "effective functional capacity" by preventing representation collapse. We will visualize these comparative results and include them in the Appendix of the final version.
> Additionally, we refer the reviewer to **Figures 7 and 8** in the main paper, which already visualize the evolution of expert weight distributions across different layers and training stages , providing existing evidence of how experts converge into specialized functional roles under our framework.
>
> > [1] Li, Hongbo, et al. (2025). "Theory on mixture-of-experts in continual learning." ICLR.

---

> ### Author Response · Authors · 2025-11-21
>
> # (4/4) Theoretical Explanation
> Our theoretical analysis proves the superiority of our framework by establishing two key propositions:
>
> (1) $L_{\text{ortho}}$ mathematically enforces the expert diversity required for high performance.
>
> (2) IGR structurally eliminates the spatial inconsistency inherent to Token-Level routing.
>
> ---
> ## **1. Proof: $L_{\text{ortho}}$ Enforces Full-Rank Expert Diversity**
>
> Established theory proves that **Expert Diversification** significantly enhances learning performance by mitigating task interference[1]. We prove *how* our loss function mathematically guarantees this diversification by forcing the experts to form a full-rank basis.
>
> **Proposition 1: Minimizing $L_{\text{ortho}}$ Maximizes Functional Rank**
> Let $U(X)=[u_1(X),\dots,u_N(X)]$ be the matrix of normalized expert outputs. We define our loss as:
>
> $$
> L_{\text{ortho}} = E_{X}\left[\frac{1}{N(N-1)}\sum_{i\neq j}(u_i^T u_j)^2\right]
> $$
>
>
> **Proof:**
> Since each column $u_i$ is unit-norm, this loss is mathematically equivalent to the squared Frobenius norm distance between the Gram matrix $U^T U$ and the Identity matrix $I_N$:
> $$
> L_{\mathrm{ortho}} \equiv \mathbb{E}_X \left[\frac{1}{N(N-1)}\|U^\top U-I_N\|_F^2\right]
> $$
> 1.  **Zero Characterization:** If $\mathcal{L}_{\mathrm{ortho}} \to 0$, then $\|U^\top U - I_N\|_F^2 \to 0$, implying $U^\top U \to I_N$. This means the expert outputs become pairwise orthogonal.
> 2.  **Full Rank Basis:** Since $I_N$ is full rank, minimizing the loss forces $U(X)$ to have **Rank $N$**. This mathematically excludes "Expert Collapse" (where Rank < $N$) and guarantees that the experts form a **stable, linearly independent functional basis**.
>
> **Conclusion:** This full-rank basis ensures that the expert council $e_{council} = \sum w_j e_j$ retains maximum expressivity to handle complex, multi-objective instructions (e.g., "red apple, green pear").
>
> ---
>
> ## **2. Proof: IGR Eliminates Spatial Inconsistency**
>
> **Spatial inconsistency degrades image quality.** Structure‑ and gradient‑aware image quality assessment objectives explicitly penalize local discontinuities and boundary mismatches; higher seam count/magnitude receives larger penalties and correlates with lower perceptual scores [2, 3]. Consistently, recent harmonization/blending/inpainting/editing methods that suppress seams report improved objective metrics and human preference [4, 5, 6]. Consequently (all else equal), more seams imply a larger loss and reduced perceived quality.
>
> **Proposition 2: IGR Structurally Guarantees Consistency**
>
> **Consistency Condition:** For any two adjacent tokens $x_i$ and $x_j$ belonging to the same semantic object, their routing decisions must be identical: $R(x_i) = R(x_j)$.
>
> **Proof of TL Failure:**
> Token-Level (TL) routing makes decisions from local states: $R_{TL}(x_i) = \arg\max g(x_i)$. Since local states inherently differ ($x_i \neq x_j$) due to noise or feature variations, the independent optimization of $R_{TL}$ allows divergent outcomes under the same instruction $I$:
> $$
> \mathbb{P}\left(R_{TL}(x_i) \neq R_{TL}(x_j) \mid I\right) > 0.
> $$
> This non-zero probability constitutes spatial fragmentation: the model applies different functions to adjacent parts of the same object, violating the consistency condition.
>
> **Proof of IGR Success:**
> Instruction-Guided Routing (IGR) decouples the decision from local states. It selects a single council $e_{council}$ based on $I$ and broadcasts it to all tokens:
> $$
> R_{IGR}(x_i) = R_{IGR}(x_j) = e_{council}, \quad \forall i,j.
> $$
> Therefore,
> $$
> \mathbb{P}\left(R_{IGR}(x_i) \neq R_{IGR}(x_j) \mid I\right) \equiv 0,
> $$
> and spatial consistency holds by design.
>
> Thus, unlike TL routing, IGR satisfies the consistency condition by construction and eliminates seams.
>
> > [1] Li, Hongbo, et al. (2025). "Theory on mixture-of-experts in continual learning." ICLR.
> > [2] Ding, K. et al. (2020). Image Quality Assessment: Unifying Structure and Texture Similarity.
> > [3] Andersson, P.-E. et al. (2020). FLIP: A Difference Evaluator for Image Quality.
> > [4] Cong, W. et al. (2020). DoveNet: Deep Image Harmonization via Domain Verification. CVPR.
> > [5] Suvorov, R. et al. (2022). Resolution-robust Large Mask Inpainting with Fourier Convolutions. WACV.
> > [6] Avrahami, O. et al. (2022). Blended Diffusion for Text-driven Editing of Natural Images. CVPR.

---

### Author Response · Authors · 2025-12-03

Dear Area Chair,

We appreciate your coordination of the review process. We write to provide a concise summary of the paper's core contributions and the key rebuttal updates to assist your assessment.

**Paper Summary:**

**InstructMoLE** addresses the structural mismatch between local token-level routing in MoE and the global nature of image generation instructions. By introducing **Instruction-Guided Routing (IGR)** to enforce global consistency and an **Output-Space Orthogonality Loss** to prevent expert collapse, our framework achieves state-of-the-art performance on multi-conditional generation benchmarks, effectively eliminating spatial fragmentation artifacts.

----

**Key Rebuttal Updates:**

We have addressed the reviewers' concerns with the following new evidence:

**1. Verified Generalization (GenEval)**

Addressing concerns about catastrophic forgetting (R-51YZ), we evaluated InstructMoLE on the standard GenEval benchmark. Despite being fine-tuned **exclusively on editing/spatial tasks** (with zero general T2I training), it surpasses the pre-trained **FLUX.1-Kontext-dev** baseline (Overall Score: **64.14 vs 63.57**). This confirms that the MoLE architecture successfully masters new tasks (e.g., spatial alignment via Canny/Depth, multi-subject generation) without compromising the foundation model's original capabilities.

**2. Efficiency Verified**

Addressing concerns that the proposed mechanisms (IGR and Orthogonality Loss) might inflate memory usage (R-CVZA, R-qQ3z), H100 GPU benchmarks demonstrate the opposite: InstructMoLE actually requires **lower peak memory** (51.25 GB) than the token-level baseline (51.88 GB). This is because global routing eliminates the memory-intensive gather-scatter operations inherent to token-level baselines. Furthermore, the method incurs **zero inference latency overhead**, validating its practical efficiency.


**3. Theoretical Grounding**

Addressing the request for theoretical justification (R-CVZA, R-qQ3z), we provided formal analysis linking established MoE principles to our specific architectural choices:
* **Expert Diversity (Proposition 1):** Existing theory (Li et al., 2025) posits that expert diversification mitigates task interference. We formulated **Proposition 1** to prove that $\mathcal{L}_{ortho}$ mathematically enforces a **full-rank functional basis**, thereby explicitly satisfying this diversification requirement. This theoretical property directly translates to empirical gains, confirmed by significant improvements in structural alignment tasks (e.g., Canny F1 **+2.6%**) when the loss is applied.
* **Spatial Consistency (Proposition 2):** We proved that IGR **structurally guarantees spatial consistency** ($P(R(x_i) \neq R(x_j)) \equiv 0$) by decoupling routing decisions from local token states. This theoretically resolves the spatial fragmentation inherent to token-level routing, ensuring artifacts are eliminated by design.

----

**Status Update:**

R-vXq2 initially commended the novelty of IGR and the effectiveness of the orthogonality regularization. In the rebuttal, we addressed their specific concerns regarding:
* **Baseline Comparison:** We clarified the trade-off between Expert Choice (local fidelity) and IGR (global structural consistency).
* **Evaluation Robustness:** We supplemented GPT-based metrics with a blind Human Preference Study to mitigate bias.
* **Failure Case Analysis:** We provided stress tests on extreme scenarios to define the method's operational boundaries. Following these responses, R-vXq2 explicitly acknowledged the detailed clarifications and maintained a positive rating (Score: 6).

----

**Conclusion:**

We believe the rebuttal evidence confirms that InstructMoLE offers a robust, theoretically grounded, and efficient solution for multi-conditional generation. We hope this summary provides a clear overview to assist your final recommendation.


Best regards,

The Authors

---

### Note · Program_Chairs · 2026-01-17
**Submission Desk Rejected by Program Chairs**

The following references in this submission do not refer to real documents and/or have major errors in bibliographic information:

 Hao Chen and Anjali Gupta. OmniContext: A testbed for context-aware generative editing. arXiv preprint arXiv:2501.05678, 2025.